# SM proteins Sly1 and Vps33 co-assemble with Sec17 and SNARE complexes to oppose SNARE disassembly by Sec18

Braden T Lobingier[1], Daniel P Nickerson[1], Sheng-Ying Lo[1], Alexey J Merz[1,2]*

[1]Department of Biochemistry, University of Washington School of Medicine, Seattle, United States; [2]Department of Physiology and Biophysics, University of Washington School of Medicine, Seattle, United States

**Abstract** Secretory and endolysosomal fusion events are driven by SNAREs and cofactors, including Sec17/α-SNAP, Sec18/NSF, and Sec1/Munc18 (SM) proteins. SMs are essential for fusion in vivo, but the basis of this requirement is enigmatic. We now report that, in addition to their established roles as fusion accelerators, SM proteins Sly1 and Vps33 directly shield SNARE complexes from Sec17- and Sec18-mediated disassembly. In vivo, wild-type Sly1 and Vps33 function are required to withstand overproduction of Sec17. In vitro, Sly1 and Vps33 impede SNARE complex disassembly by Sec18 and ATP. Unexpectedly, Sec17 directly promotes selective loading of Sly1 and Vps33 onto cognate SNARE complexes. A large thermodynamic barrier limits SM binding, implying that significant conformational rearrangements are involved. In a working model, Sec17 and SMs accelerate fusion mediated by cognate SNARE complexes and protect them from NSF-mediated disassembly, while mis-assembled or non-cognate SNARE complexes are eliminated through kinetic proofreading by Sec18.

*For correspondence: merza@uw.edu

## Introduction

Membrane fusion, the final stage of intracellular vesicular traffic, is tightly regulated so that cargos are delivered to destination compartments in an accurate and timely manner (*Bonifacino and Glick, 2004*). The core proteins required for fusion are conserved from the yeast vacuole to the synaptic active zone (*Table 1*). These include compartment-specific SNARE and SM (Sec1/Munc18) proteins, the SNARE disassembly ATPase Sec18/NSF (N-ethylmaleimide-sensitive factor), and its essential recruitment adapter Sec17/α-SNAP (soluble NSF attachment protein; *Jackson and Chapman, 2008*; *Jahn and Fasshauer, 2012*; *Südhof and Rothman, 2009*; *Ungar and Hughson, 2003*; *Wickner and Schekman, 2008*).

During docking, SNAREs on apposed vesicle and target membranes oligomerize *in trans*, 'zippering' into an ultrastable coiled-coil bundle. SNARE zippering pulls the membranes into tight apposition, locally deforming and dehydrating the bilayers to initiate fusion and compartmental mixing (*Hanson et al., 1997*; *Nichols et al., 1997*; *Sutton et al., 1998*; *Fasshauer et al., 2002*). Following fusion, individual SNAREs are entrapped within stable, fusion-inactive *cis*-complexes. To separate the SNAREs and energize them for additional instances of *trans*-complex assembly and membrane fusion, Sec17 binds the *cis*-SNARE complex, in turn recruiting Sec18. Sec18, a hexameric AAA-family ATPase, disassembles the *cis*-SNARE complex and ejects Sec17 (*Sollner et al., 1993*; *Mayer et al., 1996*; *Hanson et al., 1997*; *Littleton et al., 1998*; *Grote et al., 2000*; *Wimmer et al., 2001*; *Marz et al., 2003*; *Cipriano et al., 2013*).

SNAREs alone can fuse membranes in vitro (*Weber et al., 1998*), but fusion in vivo requires additional cofactors including regulatory small G proteins, compartment-specific tethers, and proteins of the SM family (*Jackson and Chapman, 2006*; *Wickner and Schekman, 2008*; *Südhof and Rothman, 2009*; *Yu and Hughson, 2010*; *Jahn and Fasshauer, 2012*). SM proteins are SNARE-interacting

**eLife digest** Eukaryotic organisms, from single-celled yeast to humans, divide their cells into membrane-bound compartments (organelles) of distinct function. To move from one compartment to another, or to enter or exit a cell, large molecules like proteins are packaged into small membrane sacs called vesicles.

To release its cargo, the membrane of a vesicle must fuse with the membrane of the correct destination compartment. The SNARE family of proteins plays a key role in this fusion process. As the membranes of a vesicle and target compartment come close, SNARE proteins located on each membrane form a SNARE complex that tethers the vesicle in place and causes the two membranes fuse. SNARE proteins do not act alone in this process: the SM family of proteins also plays an essential role in SNARE-mediated membrane fusion. However, it is still not clear exactly why the SM proteins are needed.

Lobingier et al. used the yeast model organism and biochemical studies with purified proteins to show that SM proteins help SNARE complexes form at the right time by regulating the delicate balance between SNARE complex formation and disassembly. This is achieved through the interplay of SM proteins and two other proteins (Sec17 and Sec18). Sec17 is known to load Sec18 onto SNARE complexes to break them apart. Lobingier et al. showed that Sec17 can also load SM proteins on SNARE complexes. This hinders Sec18 action, and so helps to keep the SNARE complexes intact. Because each SM protein tested only binds to the SNARE complex that should function at the membrane where the SM protein resides, these findings suggest SM proteins perform quality control at potential sites of membrane fusion.

~600 residue proteins with a highly conserved tertiary fold (*Carr and Rizo, 2010*; *Rizo and Südhof, 2012*). Four SM subfamilies are essential for fusion within specific subcellular domains: ER and Golgi (Sly1; *Cao and Barlowe, 2000*; *Dascher et al., 1991*); plasma membrane (Sec1/Munc18; *Grote et al., 2000*; *Harrison et al., 1994*; *Hosono et al., 1992*; *Novick et al., 1981*; *Verhage et al., 2000*; *Weimer et al., 2003*); endosomes (Vps45; *Cowles et al., 1994*; *Piper et al., 1994*); and late endolysosomal organelles (Vps33; *Banta et al., 1990*; *Wada et al., 1990*). The in vivo SM requirement is so general and so stringent that SMs are now considered, along with SNAREs, to be components of the core fusion machinery (*Südhof and Rothman, 2009*). However, the biochemical mechanisms underlying the in vivo SM requirement are opaque.

Various hypotheses have been proposed to explain the function of SMs in fusion. A major reason for the proliferation of models is that different SMs have divergent SNARE binding modalities (*Carr and Rizo, 2010*; *Rizo and Südhof, 2012*). However, accruing evidence suggests that SMs share a core ability to bind cognate ternary or quaternary SNARE bundles (*Carr et al., 1999*; *Scott et al., 2004*; *Carpp et al., 2006*; *Togneri et al., 2006*; *Dulubova et al., 2007*; *Kramer and Ungermann, 2011*; *Lobingier and Merz, 2012*). These observations prompted the conjecture that the central, evolutionarily conserved function of SM proteins involves their direct association with assembling pre-fusion *trans*-SNARE complexes (*Carr and Rizo, 2010*; *Rizo and Südhof, 2012*). Indeed, Sec1 and Munc18-1 accelerate SNARE-mediated liposome fusion by several-fold (*Scott et al., 2004*; *Shen et al., 2007*; *Rathore et al., 2010*). This acceleration is contingent on initial reaction conditions, and recent experiments show that Munc18-1 stimulates liposome fusion more efficiently in concert with the specialist exocytosis cofactors Munc13 and synaptotagmin (*Ma et al., 2013*). Similarly, Vps33, Vps45, and Sly1 accelerate liposome fusion, but with nearly absolute requirements for additional factors including Rab proteins and tethering factors (*Hickey et al., 2009*; *Ohya et al., 2009*; *Furukawa and Mima, 2014*). It remains unclear how SM proteins accelerate SNARE-mediated fusion, and it is unknown whether the kinetic stimulation observed in vitro is sufficient to explain absolute requirements for SMs in vivo.

In a different and not mutually exclusive role, SM proteins might interact with SNARE recycling factors. In vitro, Sec18 and Sec17 can disassemble pre-fusion *trans*-SNARE complexes and can prevent the fusion of intact yeast lysosomal vacuoles or liposomes (*Ungermann et al., 1998*; *Rohde et al., 2003*; *Mima et al., 2008*; *Stroupe et al., 2009*). Premature SNARE disassembly was impeded by the Vps-C tethering complex HOPS (*Xu et al., 2010*). This protective activity of HOPS was hypothesized to reside within its SM subunit Vps33, which is necessary and sufficient for HOPS binding to the

**Table 1.** Nomenclature of general and compartment-specific SNAREs and SNARE cofactors employed in this study, and their equivalents in mammalian synaptic exocytosis

|  | Yeast | Mammal |
|---|---|---|
|  | General | General |
| AAA-family ATPase | Sec18 | NSF |
| Sec18 adapter | Sec17 | α-SNAP |

|  | Golgi | Vacuole | Synaptic exocytosis |
|---|---|---|---|
| SM protein | Sly1 | Vps33 | Munc18-1 |
| Qa-SNARE | Sed5 | Vam3 | Syntaxin |
| Qb-SNARE | Bos1 | Vti1 | SNAP-25 (N-domain) |
| Qc-SNARE | Bet1 | Vam7 | SNAP-25 (C-domain) |
| R-SNARE | Sec22 | Nyv1 | Synaptobrevin (VAMP2) |

The Q/R taxonomy of SNARE domain subfamilies is derived from *Fasshauer et al. (1998)*.

vacuole SNARE complex (*Lobingier and Merz, 2012*). Similarly, Munc18-1, acting in concert with Munc13 and synaptotagmin, facilitated SNARE-mediated liposome fusion in the presence of otherwise inhibitory concentrations of NSF and α-SNAP (Sec18 and Sec17; *Ma et al., 2013*). In the absence of NSF and α-SNAP, Munc18 and Munc13 had little or no effect on the extent of SNARE-mediated liposome fusion when compared to fusion driven solely by SNAREs and synaptotagmin. These findings led to proposals that SMs protect pre-fusion SNARE complexes from premature disassembly while exposing post-fusion complexes and mis-assembled or non-cognate pre-fusion complexes to disassembly by Sec18 (*Mima et al., 2008*; *Starai et al., 2008*; *Rizo and Südhof, 2012*). However, central predictions of these models are still untested. Direct protection of a SNARE complex by an SM has not been experimentally demonstrated, and it is unknown whether SM proteins functionally interact (or compete) with the disassembly machinery in living cells.

Using *Saccharomyces cerevisiae* as an experimental platform, we tested the hypothesis that SMs functionally interact not only with SNAREs, but also with Sec17 and Sec18. Through a combination of genetic manipulations in vivo, and in vitro assays of SNARE complex assembly and disassembly, we establish that SM proteins directly impair Sec18-mediated SNARE disassembly. In the course of these studies we discovered that Sec17 directly promotes selective loading of at least two different SM proteins onto cognate SNARE complexes. Moreover, an extraordinarily steep temperature dependence limits SM loading onto SNARE complexes, implying that SM-SNARE complex formation entails significant conformational transitions. The thermal dependence of SM loading may partially explain why SNARE–Sec17–SM complexes eluded detection in previous studies.

## Results

### Wild-type SM proteins are required to resist Sec17 and Sec18 overproduction

In vitro reconstitution experiments led to models in which SM proteins, in conjunction with additional SNARE cofactors, functionally oppose Sec17 and Sec18 activity (*Mima et al., 2008*; *Stroupe et al., 2009*; *Xu et al., 2010*; *Ma et al., 2013*). To probe for antagonism between SMs and SNARE disassembly factors in vivo, we turned to the late endolysosomal SM Vps33. We recently characterized a hypomorphic *VPS33* allele, *vps33*$^{car}$. Vps33$^{car}$ (G297V) is an analog of the *Drosophila* Vps33a (G249V) mutant, encoded by *carnation*[1], probably the first SM allele ever isolated (*Patterson, 1932*; *Sevrioukov et al., 1999*). Null mutant *vps33Δ* cells, or *vps33*$^{R281A}$ functional nulls (*Lobingier and Merz, 2012*), have severe trafficking defects, lack identifiable vacuolar lysosomes (*vps* class C morphology; *Raymond et al., 1992*), and are inviable at 37°C. In contrast, *vps33*$^{car}$ mutants retain partial function, with milder defects in vacuolar cargo sorting, moderate (Class B) vacuole fragmentation, and slow growth at 37°C (*Lobingier and Merz, 2012*). When Sec17 and Sec18 were overproduced in wild-type *VPS33* cells, growth was normal at either standard temperature (30°C) or at 37°C (*Figure 1A*). In marked contrast, Sec17 and Sec18 overproduction in mutant *vps33*$^{car}$ cells caused severe growth defects at 37°C (*Figure 1A*). As an additional control we overproduced Sec17 and Sec18 in another Class B *vps* mutant, *vps41Δ*. Vps41 is, with Vps33, a subunit of the HOPS tethering complex. There was little or no growth defect when Sec17 and Sec18 were overproduced in *vps41Δ* cells (*Figure 1A*).

The late endosome and vacuolar lysosome are required for metal tolerance in *S. cerevisiae*. For this reason, growth in the presence of added Zn$^{2+}$ is a classical indicator of intact endolysosomal function.

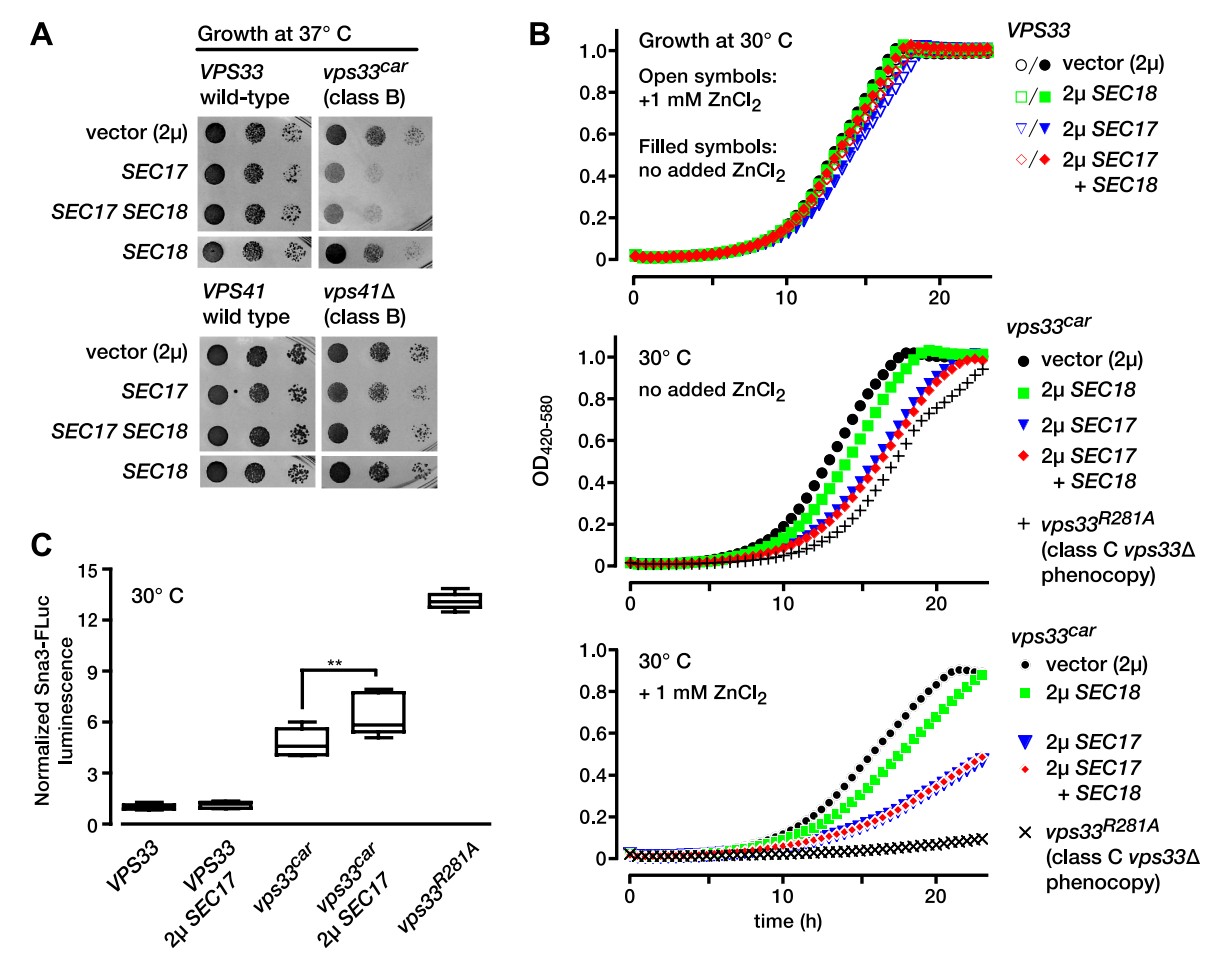

**Figure 1**. Partial Vps33 deficiency sensitizes cells to overproduction of SNARE disassembly proteins. (**A**) Limiting dilution growth assay on synthetic media agar plates incubated at 37°C. (**B**) Growth curves in selective, synthetic liquid media (YNB lacking uracil and containing 0.05% casamino acids and 2% dextrose, with or without 1 mM ZnCl₂). Data points represent the means of n = 4 samples. (**C**) LUCID analysis of luminal sorting efficiency of the vacuole cargo Sna3-fLuc. Note that *vps33car* is a hypomorphic allele with partial loss-of-function, while *vps33 R281A* is a functional null with total loss of function. Box plots summarize n = 5 biological replicates, except for *vps33 R281A*
(n = 4). **p<0.01 (one-way ANOVA). fLuc, firefly luciferase. *2μ*, high copy plasmid vector.

Wild-type *VPS33* cells grew at equal rates with or without 1 mM Zn²⁺. Overproduction of Sec17, Sec18, or both together had almost no effect on growth of *VPS33* cells, with or without added Zn²⁺ (*Figure 1B*, top panel). *vps33car* mutants grew almost as well as *VPS33* cells, but when Sec17, or Sec17 and Sec18 were overproduced together, the *vps33car* mutants grew slowly (*Figure 1B*, middle panel), a defect markedly enhanced by 1 mM Zn²⁺ (*Figure 1B*, bottom panel). Thus, Sec17 or Sec17 and Sec18 over-production strongly exacerbate defects in endolysosomal function, even when Vps33 function is only partially impaired. To test for synthetic trafficking defects we employed LUCID, a quantitative assay of traffic from the Golgi to the late endosome. LUCID uses a chimeric reporter, the cargo protein Sna3 fused to firefly luciferase (fLuc). Sna3-fLuc accumulates when endolysosomal traffic or cargo sorting into multivesicular bodies is impaired (*Nickerson et al., 2012*; *Paulsel et al., 2013*). *Renilla* luciferase is co-expressed to control for expression and nonspecific protein turnover. Sec17 overproduction in *vps33car* cells, but not in wild-type cells, significantly impaired Sna3-fLuc sorting (*Figure 1C*). Together, the data show that full Vps33 function is required to withstand either Sec17 or Sec17 and Sec18 overproduction.

To test for functional interactions between Sec17 and Sec18 and another SM, we studied a conditional mutant of Sly1, the Golgi SM. *sly1ts* mutant cells grow almost as well as wild-type cells at

permissive temperature (26°C) but cannot grow at elevated temperatures (*Cao and Barlowe, 2000*). Wild-type *SLY1* cells grew normally when Sec17 and Sec18 were overproduced, alone or together (*Figure 2*). In contrast, overproduction of Sec17, or Sec17 and Sec18 together, profoundly impaired the growth of *sly1*[ts] mutants, even at permissive temperatures (24–26°C; *Figure 2*). In independent work, *sly1*[ts] *sec18-1* double mutants were inviable (*Kosodo et al., 2003*). We conclude that full, wild-type Sly1 and Vps33 function is required to buffer cells against perturbations of the SNARE disassembly machinery.

## SM proteins reduce the rate of SNARE disassembly by Sec18

To test the hypothesis that SM proteins directly regulate the activities of Sec17 and Sec18, we established an in vitro assay of SNARE complex disassembly (*Figure 3A*). Vacuole and Golgi SM proteins, cognate SNAREs, and Sec17 and Sec18 were individually purified (*Table 1*; *Figure 3—figure supplement 1*). Golgi or vacuole SNARE complexes (*Table 1*) were then assembled on immobilized Qa-SNAREs. We emphasize that the SNARE constructs, assembled on affinity supports to probe protein–protein interactions, encoded only cytoplasmic domains, not transmembrane segments. Consequently, the complexes formed from these proteins cannot be described using the membrane-dependent topological terms *cis*- and *trans*-. Using conditions optimized for Vps33–SNARE complex binding (*Lobingier and Merz, 2012*), SNARE complexes were incubated with Sec17, with or without the cognate SM (Vps33 or Sly1). Sec18 was then added to initiate disassembly. In the absence of SMs, Sec18 rapidly and completely disassembled the SNARE complexes (*Figure 3B,C*, lanes 1–5). Disassembly required ATP and was blocked when Mg²⁺ was sequestered by EDTA (*Figure 3B,C*, compare lanes 5 and 6). Pre-incubation with SMs (Sly1 or Vps33) delayed, but did not prevent, SNARE disassembly (*Figure 3B,C*, lanes 7–11). The effect of Sly1 was quantified: pre-incubation with the Sly1 decreased the rate of disassembly by $63 \pm 9\%$ (*Figure 3—figure supplement 2*). The Sed5 (Qa-SNARE) used for the experiments shown in *Figure 3B,D* contained only the SNARE domain, and not the H$_{abc}$ or N-peptide domains. However, Sly1 also protected Golgi SNARE complexes assembled on Sed5 full cytoplasmic domain rather than Sed5 SNARE domain (*Figure 3—figure supplement 3*). In control

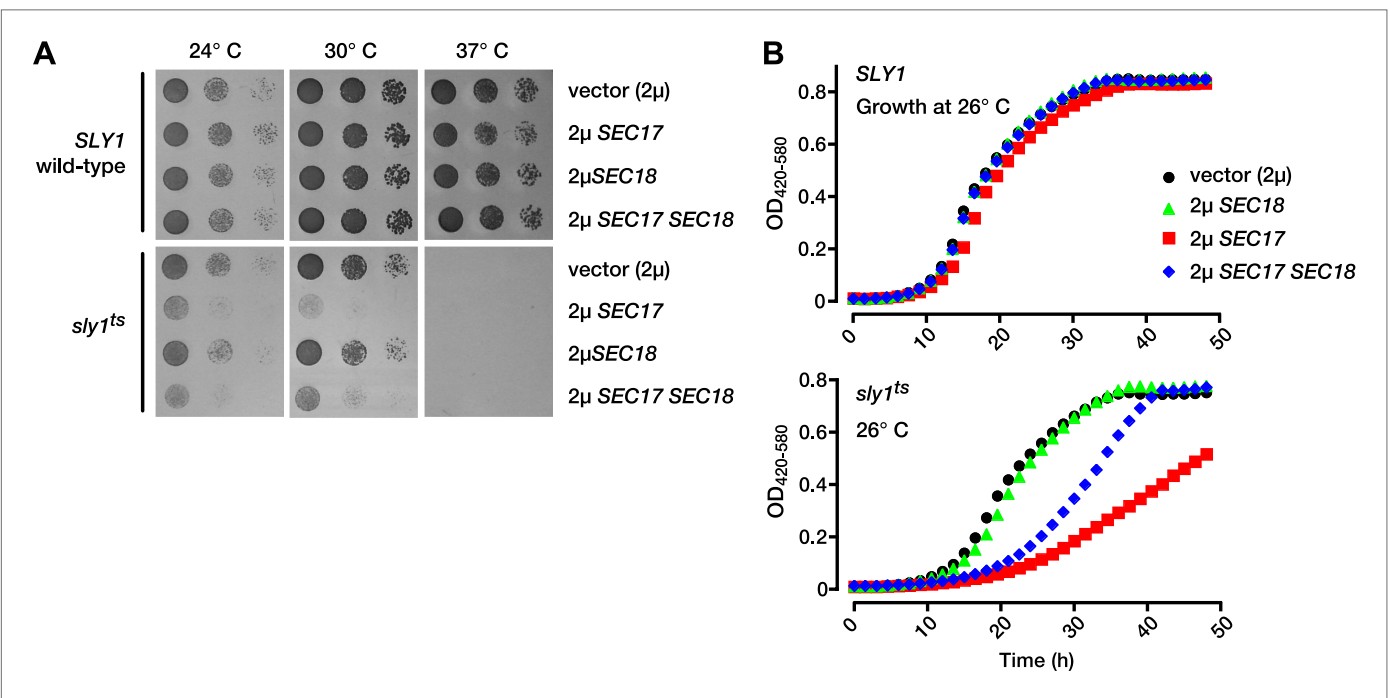

**Figure 2**. Partial Sly1 deficiency sensitizes cells to overproduction of SNARE disassembly proteins. (**A**) Limiting dilution growth assay on plasmid-selective, synthetic media agar plates at 24°, 30° and 37°C. (**B**) Growth curves of yeast in selective, synthetic liquid media at 26°C. Data points each represent the mean of nine replicate samples. *2μ*, high copy plasmid vector.

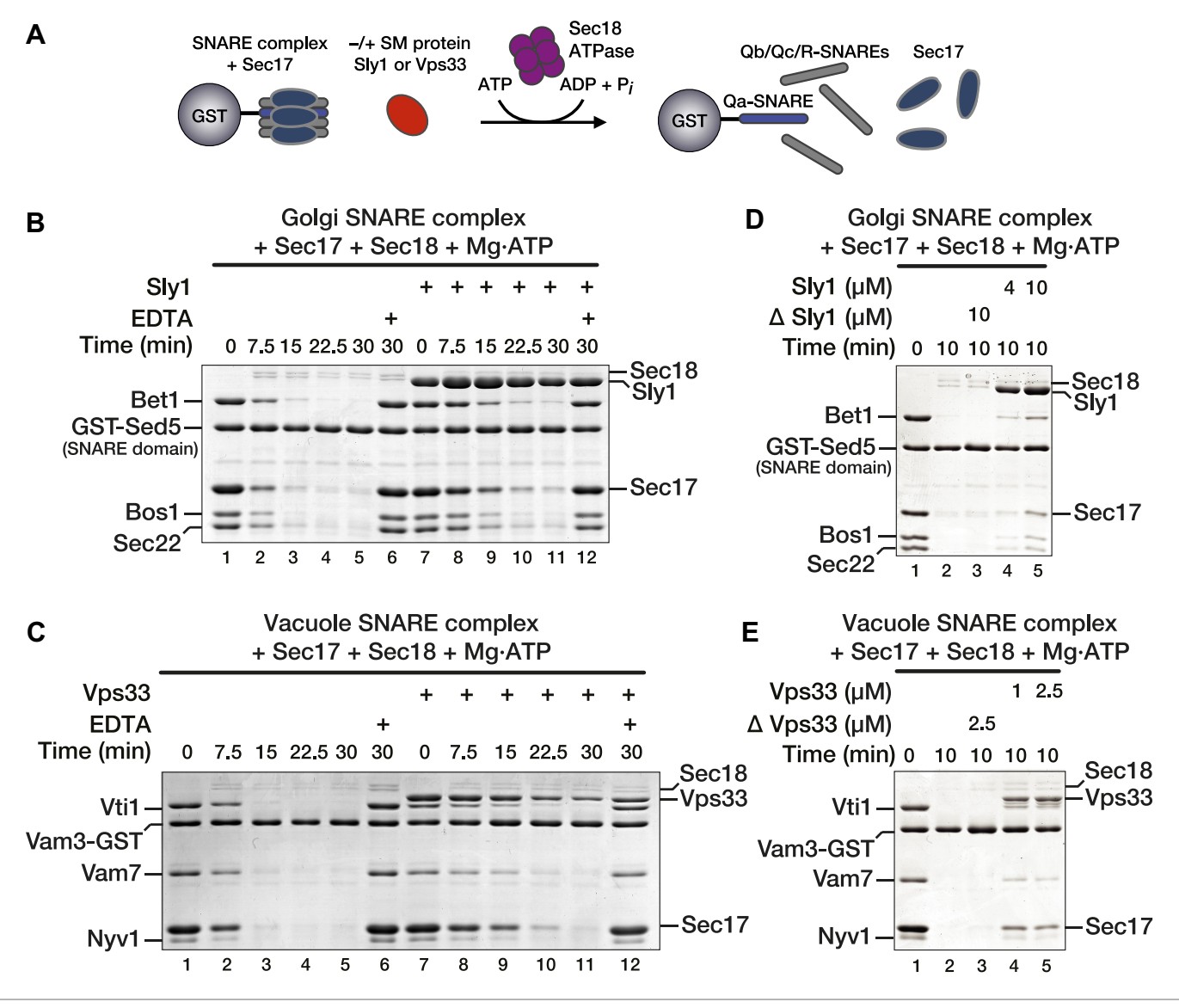

Figure 3. SM proteins oppose Sec18-mediated SNARE disassembly. (A) Schematic of SNARE disassembly assay. SNARE complexes assembled onto immobilized Qa-SNAREs (Vam3 cytoplasmic domain or Sed5 SNARE domain) were pre-incubated in the presence of Sec17, with or without added SM (Vps33 or Sly1). Sec18 was added to initiate disassembly. The remaining resin-bound material was washed, collected, and analyzed by SDS-PAGE at indicated times. (A and B) Sec17 (20 μM) and Sly1 (10 μM) or Vps33 (2.5 μM) were pre-incubated with SNARE complexes (500 nM) at 30°C for 60 min in Disassembly Buffer. Sec18 (300 nM) was then added. Under these conditions each Sec18 hexamer catalyzed disassembly of >10 SNARE complexes. In negative controls (lanes 6 and 12), $Mg^{2+}$ was chelated with EDTA prior to Sec18 addition. (D and E) SNARE complex disassembly was assayed as in B and C, but with variable SM protein concentrations as indicated. In ΔSly1 or ΔVps33 lanes, the SM solutions were heated in a boiling water bath for 10 min, plunged into ice-water, and then clarified at 20 k × g to rule out effects of potential heat-stable contaminants in the preparations.

The following figure supplements are available for figure 3:

Figure supplement 1. Purified Proteins.

Figure supplement 2. Quantification of SNARE complex protection by Sly1.

Figure supplement 3. Sly1 protection of SNARE complexes assembled on Sed5 containing Habc domain and N-peptide.

Figure supplement 4. Pre-incubation of Sly1 with SNARE complexes increases the fraction of SNARE complex resistant to Sec18-mediated disassembly.

reactions the ability of the Sly1 and Vps33 preparations to impede SNARE disassembly was heat labile (**Figure 3D,E**, compare lanes 3 and 5). Omitting the pre-incubation step reduced both SM binding and SNARE complex protection (**Figure 3—figure supplement 4**).

Substantial amounts of Vps33 and Sly1 remained bound to immobilized Qa-SNAREs even after complete SNARE complex disassembly (**Figure 3B,C**, lanes 11). This raised the possibility that the SMs could capture Qa-SNAREs in an assembly-active state. Thus, it was necessary to test whether bound SM proteins reduce the rate of SNARE complex disassembly or, alternatively, accelerate SNARE complex re-assembly. To evaluate these alternatives, SNARE complexes were completely disassembled using Sec17, Sec18, and ATP. Sec18 activity was then quenched with EDTA and the reactions were incubated for an additional 30 min (**Figure 4A**). At the low concentrations of free SNAREs liberated by disassembly, no re-assembly of SNARE complexes was detected within 30 min in either the absence or presence of SM (**Figure 4B,C**, compare lanes 1 and 3, and lanes 2 and 4). We next tested the competence of the Qa-SNARE for de novo assembly following Sec18-mediated disassembly. After complete disassembly, additional Qb, Qc, and R-SNAREs were added along with the EDTA quench (**Figure 4B,C**, compare lanes 5 and 6). Under these conditions re-assembly occurred, but the rates of re-assembly were not increased by either Vps33 or Sly1, in accord with previous reports that these SMs do not accelerate SNARE assembly in solution (**Kosodo et al., 2002**; **Peng and Gallwitz, 2002**; **Hickey and Wickner, 2010**).

We conclude that Vps33 and Sly1 kinetically impair, but do not prevent, Sec18-mediated SNARE disassembly. Because Vps33 is both necessary and sufficient for HOPS binding to SNARE core bundles (**Lobingier and Merz, 2012**), Vps33 likely accounts for the ability of HOPS to shield *trans*-SNARE complexes from premature disassembly (**Xu et al., 2010**). Because Vps33 and Sly1 exhibit similar activities, protection of *trans*-SNARE complexes from Sec18/NSF may be a more general feature of

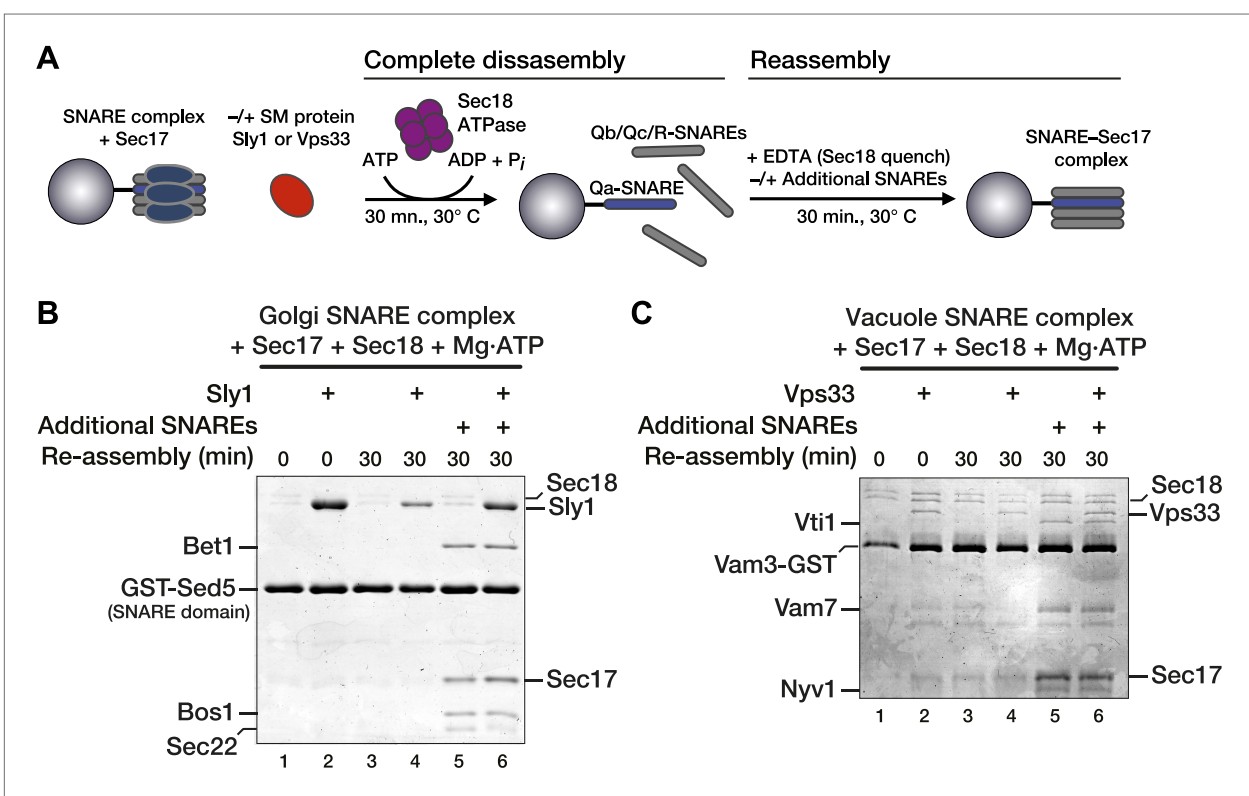

**Figure 4**. Vps33 and Sly1 do not accelerate SNARE complex re-assembly in solution. (**A**) Cartoon schematic of the re-assembly assay. (**B** and **C**) SNARE complexes (500 nM) were assembled as in **Figure 3**. Following pre-incubation of SNARE complexes with 20 µM Sec17 and SM (10 µM Sly1 or 2.5 µM Vps33, as indicated), SNARE complexes were completely disassembled for 30 min by Sec18. Disassembly was terminated with EDTA, and SNARE complex re-assembly was assayed after a further 30 min at 30°C. As indicated, some re-assembly reactions (lanes 5 and 6) were supplemented with soluble SNAREs (Qb, Qc, and R; ~3 µM each), which were added along with the EDTA quench.

SM biochemistry. Liposome fusion experiments with SMs from the other two SM subfamilies, Vps45 and Munc18-1, are also consistent with this interpretation (*Ohya et al., 2009*; *Ma et al., 2013*).

## Sec17 promotes Vps33 binding to the vacuole SNARE complex

Up to three copies of Sec17/α-SNAP bind per SNARE complex bundle (*Hanson et al., 1997*; *Fleming et al., 1998*; *Marz et al., 2003*; *Vivona et al., 2013*), and α-SNAP can competitively displace the presynaptic $Ca^{2+}$ sensor synaptotagmin from neuronal SNARE complexes (*Sollner et al., 1993*). Thus, it was surprising to observe both Sec17 and Vps33 bound to SNARE complexes in our disassembly assays (e.g., *Figure 3C*, lanes 7 and 12). Similarly, both Sec17 and Sly1 associated with Golgi SNARE complex, even though the Sed5 (Qa-SNARE) construct used lacked the N-peptide, a high-affinity recruitment site for the Sly1 (e.g., *Figure 3B*, lanes 7 and 12). To test whether Sec17 and SMs compete for binding, immobilized vacuolar SNARE complexes were assayed for binding of Vps33, Sec17, or both. As we previously reported (*Lobingier and Merz, 2012*), Vps33 binds the vacuole SNARE complex with low µM affinity (*Figure 5A*, lane 2). But rather than competing, Sec17 addition strongly stimulated Vps33 binding to the SNARE complex (*Figure 5A*, lane 4). Importantly, the stoichiometry of Sec17 binding was unaltered when Vps33 was also bound (*Figure 5A*, compare lanes 3 and 4).

Sec17 stimulation of Vps33 binding to the SNARE complex depended on the Sec17 concentration and tracked with Sec17 occupancy on the complex (*Figure 5B*). Vps33-SNARE complex binding was saturable (*Figure 5C*), and Sec17 increased the apparent affinity of Vps33 for SNARE complex by more than 5-fold (from $K_{D(obs)}$ = 1.60 ± 0.10 µM to 0.30 ± 0.01 µM). As these are non-equilibrium measurements, we caution that they may systematically underestimate absolute SNARE-SM affinities. Under saturation binding conditions, Sec17–Vps33–SNARE complexes assembled in apparent 1:3:1 stoichiometry (*Figure 5D*).

The above results argue that SNARE-bound Sec17 stimulates Vps33 binding. To rule out an alternative possibility, that free Sec17 in solution enhances Vps33 binding activity, Sec17 was bound to SNARE complexes, and unbound Sec17 was washed out before Vps33 was added (*Figure 5E*). Vps33 bound to SNARE complex equally well in the presence of free-plus-bound Sec17 (*Figure 5E*, lane 2) or to SNARE–Sec17 complex from which excess unbound Sec17 had been removed (*Figure 5E*, lane 3). Thus, Vps33 binding is stimulated by Sec17 on the SNARE complex, not by Sec17 in solution. In a further control, addition of bovine serum albumin (BSA) in place of Sec17 had no effect on Vps33 binding to SNARE complex (*Figure 5E*, lane 4). To test whether Vps33 binds Sec17 directly, GST-Vps33 was immobilized and assayed for binding of soluble SNARE complex, Sec17, or both (*Figure 5F*). SNARE complex and Sec17 bound efficiently to immobilized Vps33 only when both were present (*Figure 5E*, compare lane 4 to lanes 2 and 3). SNARE complexes and Sec17 therefore bind Vps33 through a cooperative mechanism involving all three components.

## Sec17 promotes Sly1 binding to the Golgi SNARE complex

A co-complex between SNAREs, Sec17, and an SM is unprecedented. Thus, it was essential to test whether similar results might be obtained with a divergent SM protein and its cognate SNAREs. We again turned to Sly1, the Golgi SM. Sly1 was previously shown to avidly bind the terminal N-peptide of the Qa-SNARE Sed5. However, N-peptide binding is dispensable for Sly1 function in vivo (*Bracher and Weissenhorn, 2002*; *Peng and Gallwitz, 2002*, *2004*). To examine N-peptide-independent Sly1 binding, Golgi SNARE complex was assembled on an immobilized, truncated Sed5 SNARE domain that lacks both the N-peptide and $H_{abc}$ domains (GST-Sed5$_{SNARE domain}$). As in previous work (*Peng and Gallwitz, 2004*) little or no Sly1 bound to the Golgi SNARE bundle. In the presence of Sec17, however, Sly1 binding was dramatically stimulated (*Figure 6A*, compare lanes 2 and 4).

As with Vps33, Sly1 binding to SNARE complex depended on the Sec17 concentration and tracked with Sec17 occupancy on the complex (*Figure 6B*, lanes 1-6). Sec17 did not associate with, or impede formation of, binary complexes between Sly1 and the N-peptide of full-length Sed5 (*Figure 6B*, lanes 7 and 8). Thus, Sec17 stimulates Sly1–SNARE complex association, but Sec17 does not efficiently bind Sly1 or Sed5 in the absence of an assembled SNARE complex. Sly1 bound to the Sec17–SNARE complex saturably and with moderate affinity (*Figure 6C*; $K_{D(obs)}$ = 1.0 ± 0.1 µM). We emphasize that this binding was measured under conditions where Sly1–Sed5 N-peptide binding cannot occur, as the Sed5 construct lacks the N-peptide and $H_{abc}$ domains. In the absence of Sec17, little or no Sly1 bound the SNARE bundle at concentrations up to 15 µM. Under saturation conditions the SNARE complex, Sec17, and Sly1 assembled in apparent 1:3:1 stoichiometry, consistent with the results for Vps33 (*Figures 5D and 6D*).

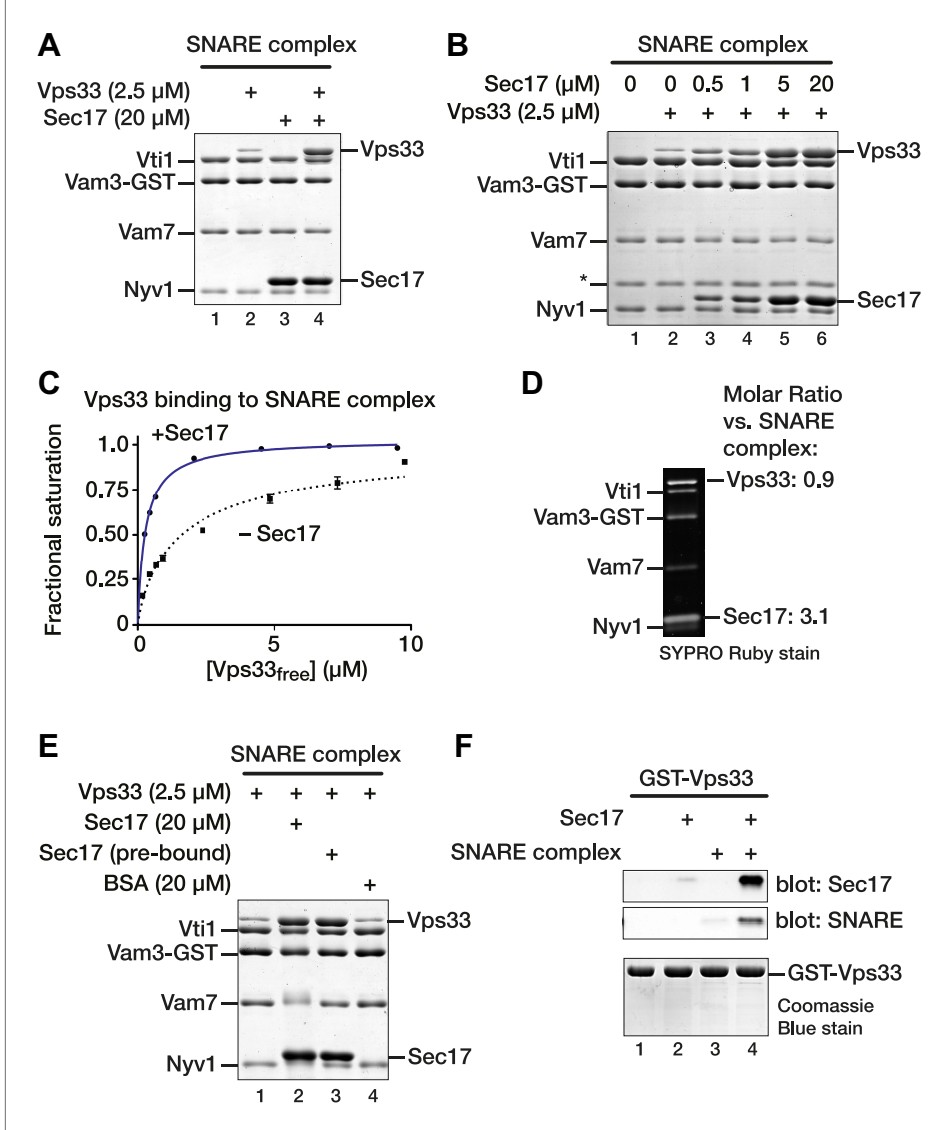

**Figure 5**. Sec17 promotes Vps33 binding to vacuole SNARE complex. (**A**) SNARE complex (500 nM) was assembled on Vam3-GST. Sec17 (20 μM), Vps33 (2.5 μM), or both were incubated with the SNARE complex for 1 hr at 30°C in SM Assay Buffer. Unbound material was washed out, then bound material was separated by SDS-PAGE and visualized with Coomassie blue. (**B**) The dose–response for Sec17 stimulation of Vps33 binding to SNARE complexes was assayed as in **A**, but Sec17 concentration was varied (0.5–20 μM) while Vps33 was held constant (2.5 μM). (**C**) Vps33 binding to SNARE complex with or without Sec17 was assayed as in **A** and **B**, except that Vps33 concentration was varied and protein bands were stained and quantified using SYPRO Ruby. The fractional saturation of total Vps33–SNARE complex binding was plotted vs free (total minus bound) Vps33. Fits of a one-site binding model yielded $Kd_{obs}$ = 300 ± 10 nM for Vps33 binding to the SNARE complex in the presence of Sec17, and $Kd_{obs}$ = 1.6 ± 0.10 μM without Sec17. Two-site or cooperative binding models did not substantially improve the fits. (**D**) To estimate the stoichiometry of SNARE–Sec17–Vps33 binding, complexes were assembled under saturation binding conditions, separated by SDS-PAGE, and analyzed using SYPRO Ruby stain. The band intensities were quantified using standard curves generated with individual purified proteins. (**E**) Vps33 binding is stimulated by SNARE-associated Sec17. Complexes were assayed in lanes 1 and 2 as in **A**. In lane 3 (Sec17 pre-bound), Sec17 was bound to SNARE complexes for 60 min at 30°C. Unbound Sec17 was washed out and 2.5 μM Vps33 was then added for an additional 60 min at 30°C. In lane 4, 20 μM BSA was substituted for Sec17. (**F**) Cooperativity of assembly. GST-Vps33 (500 nM) was immobilized and incubated with 20 μM Sec17, soluble SNARE complex, or both. Bound material was separated by SDS-PAGE and stained with Coomassie blue, or analyzed by immunoblot.

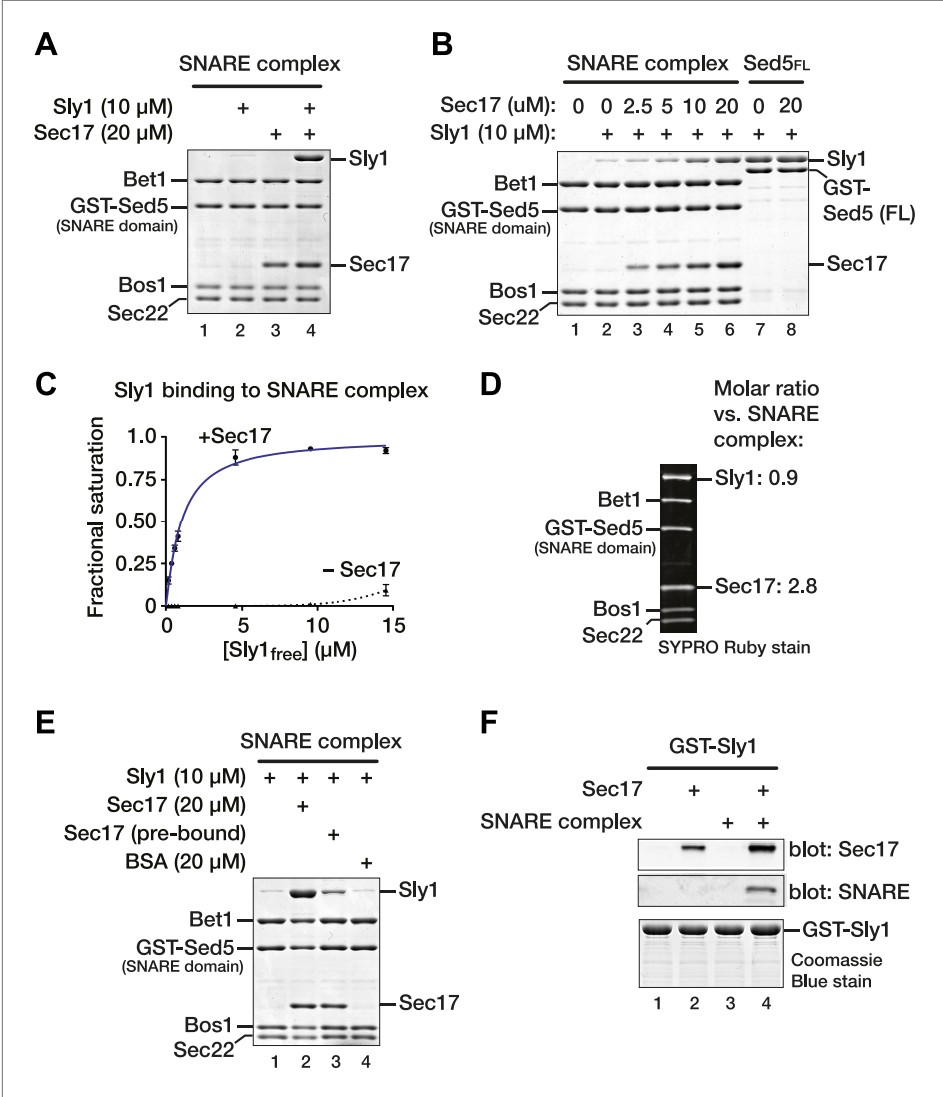

**Figure 6**. Sec17 promotes Sly1 binding to Golgi SNARE complex. (**A**) Golgi SNARE complex (500 nM) was assembled on immobilized Sed5 SNARE domain (GST-Sed5$_{SNARE\ domain}$) lacking the N-peptide and H$_{abc}$ segments. SNARE complexes were incubated with Sec17 (20 μM), Sly1 (10 μM), or both for 60 min at 30°C. Unbound proteins were washed out, and bound proteins were separated by SDS-PAGE and stained with Coomassie blue. (**B**) The dose–response for Sec17 stimulation of Sly1 binding to SNARE complexes was assayed as in **A**, but Sec17 concentration was varied (2.5–20 μM) while Sly1 was held constant (10 μM). In lanes 7 and 8, the full cytoplasmic domain of Sed5 (Sed5$_{FL}$, including the H$_{abc}$ and N-peptide segments; 500 nM) was immobilized to test for direct binding of Sec17 to Sed5 and Sly1 in the absence of assembled SNARE complex. (**C**) Sly1 binding to SNARE complex with or without Sec17 was assayed as in **A** and **B**, except that Sly1 concentration was varied and protein bands were stained and quantified using SYPRO Ruby. The fractional saturation of total Sly1-SNARE complex binding was plotted vs free (total minus bound) Sly1. Fits of a one-site binding model yielded with an apparent Kd$_{obs}$ = 1.0 ± 0.1 μM for Sly1 binding to the SNARE complex in the presence of Sec17. It was not possible to fit the no-Sec17 condition. Two-site or cooperative binding models did not substantially improve the fits. (**D**) Stoichiometry of SNARE-Sec17-Sly1 complexes assembled under saturation conditions was estimated using standard curves of purified proteins of known concentrations. (**E**) Sly1 binding is stimulated by SNARE-associated Sec17. Binding was assayed in lanes 1 and 2 as in **A**. In lane 3, Sec17 was pre-bound to SNARE complexes for 60 min at 30°C. Unbound Sec17 was then washed out and Sly1 (10 μM) was added for an additional 60 min at 30°C. In lane, 4 BSA (20 μM) was substituted for Sec17. (**F**) Cooperativity of assembly. GST-Sly1 (500 nM) was immobilized and incubated with 20 μM Sec17, SNARE complex, or both. Bound material was separated by SDS-PAGE and stained with Coomassie blue or analyzed by immunoblot.

As with Vps33, Sly1 binding was promoted by SNARE-bound Sec17 rather than Sec17 in solution (*Figure 6E*). The Sec17 concentration required to saturate Golgi SNARE complex was about four-fold greater than for vacuole SNARE complex (compare the Sec17 curves in *Figures 5B and 6B*). Consistent with this observation, Golgi SNARE complex pre-bound to Sec17 and then washed retained less Sec17, and commensurately less Sly1, versus pulldowns in which Sec17 was in excess (*Figure 6E*, compare lanes 2 and 3). No increase in Sly1 binding to Golgi SNARE complexes was observed when BSA was added instead of Sec17 (*Figure 6E*, lane 4). Sly1, like Vps33, binds SNARE–Sec17 complexes through a cooperative mechanism. GST-tagged Sly1 was immobilized and assayed for binding of soluble SNARE complex, Sec17, or both (*Figure 6F*). SNARE complex was retained by GST-Sly1 only in the presence of Sec17, and Sec17 was retained most efficiently on GST-Sly1 when SNARE complexes were present. There may be some affinity between GST-Sly1 and Sec17 in the absence of SNARE complex under these conditions. However, Sec17 did not detectably interact with Sly1 when Sly1 was tethered to the GST-Sed5 N-peptide (*Figure 6B*, lanes 7 and 8). We conclude that Sec17 promotes Sly1 and Vps33 loading onto SNARE complexes through a cooperative mechanism. The resulting assemblies contain three Sec17 molecules and one SM per quaternary SNARE bundle.

## Sec17 promotes selective SM loading onto cognate SNARE complexes

Sec17 and its mammalian homolog α-SNAP are capable of engaging all SNARE complexes. In contrast, the SM proteins operate at specific organelles and preferentially recognize cognate, pathway-specific SNAREs and SNARE complexes (*Carr and Rizo, 2010*; *Rizo and Südhof, 2012*). Sec17 can bind SNARE complexes as a trimer, raising the possibility that Vps33 and Sly1 recognize composite features of the Sec17 multimer surface but do not touch the underlying SNAREs. If this model is correct, Sec17-stimulated SM binding to SNARE complexes should exhibit no selectivity for the underlying SNARE bundle. In an alternative model, SM proteins touch and recognize cognate SNARE complexes during Sec17-stimulated binding. This model predicts that Sec17-stimulated SM binding should be SNARE-selective and compartment-specific.

To evaluate these models, Golgi and vacuole SNARE complexes were assembled and assayed for Vps33 and Sly1 binding in the absence or presence of Sec17 (*Figure 7*). In the presence of Sec17, Vps33 preferentially bound to vacuole SNARE complexes (*Figure 7A*, lanes 3 and 3'). In the reciprocal experiment, Sec17 promoted selective Sly1 binding to cognate Golgi SNARE complexes (*Figure 7B*, lanes 3 and 3'). Sec17 therefore promotes selective recognition of cognate SNARE complexes by both Vps33 and Sly1. These findings further underscore the cooperativity of Sec17,

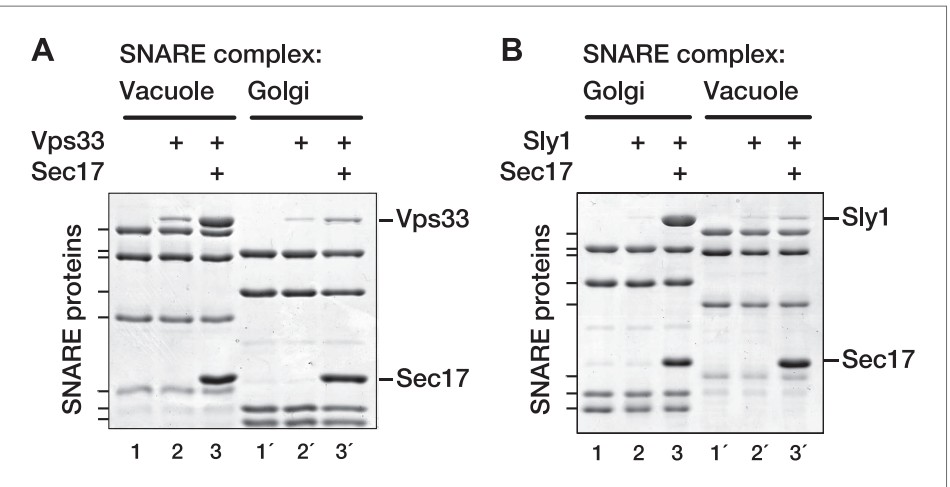

**Figure 7**. SM proteins touch and recognize cognate SNARE-Sec17 complexes. Golgi and vacuole SNARE complexes (500 nM) were assembled on affinity supports and assayed for binding in the absence or presence of Sec17 (20 µM). (**A**) Assay of Vps33 (2.5 µM) binding. (**B**) Assay of Sly1 (10 µM) binding. Golgi complexes were assembled on Sed5 SNARE domain lacking the N-peptide and $H_{abc}$ segments. Binding reactions were incubated 2 hr at 30°C, unbound material was washed out, and the bound proteins were analyzed by SDS-PAGE and Coomassie blue staining.

SM, and SNARE complex co-assembly (*Figures 5 and 6*) and indicate that even when a Sec17 trimer is bound, Vps33 and Sly1 recognize organelle-specific determinants on the underlying SNARE complex.

## SM binding to SNARE complexes is strongly temperature-dependent

Unexpectedly, Sly1 binding to the SNARE-Sec17 complex increased markedly from 4° to 30°C, the physiological growth temperature (*Figure 8A*, compare lanes 4, 7, and 10). In assays performed across a range of temperatures (*Figure 8B*), Sly1 loading onto SNARE–Sec17 complex decreased by an order of magnitude as temperature dropped from 26.5°C to 20°C. In control experiments there was no comparable temperature dependence (*Figure 8B*: binding of Sly1 to Sed5 N-peptide, and binding of Sec17 to SNARE complex). Below, we show that thermal stimulation of Sly1–SNARE complex binding is dominated by changes in association rather than dissociation kinetics. Most biochemical processes have a thermal coefficient $Q_{10}$ of 2–4 (change in activity over 10°C temperature gradient; see Hille, 1991). Sly1 binding to the Sec17-SNARE complex has an extrapolated $Q_{10}$ of ~30, indicating that the Sly1 binding mechanism entails traversal of a large thermodynamic barrier.

As with Sly1, Vps33 binding increased with temperature (*Figure 8C*, compare lanes 4, 7, and 10, and *Figure 8D*, blue triangles). In contrast, Sec17 efficiently bound the vacuole SNARE complex at 4°C. Because Vps33 has moderate affinity for vacuole SNARE complex (*Figure 5C*; *Lobingier and Merz, 2012*), it was also possible to assess the temperature sensitivity of SNARE complex–Vps33 binding without Sec17. Although Sec17 stimulated Vps33 binding (*Figure 8D*, blue triangles), Vps33 bound the SNARE complex most efficiently at elevated (physiological) temperatures even when no Sec17 was present (*Figure 8D*, red circles). Thus, elevated temperature directly enhances Vps33 association with its SNARE complex.

To evaluate whether SM binding efficiency is limited by on- or off-rates, SNARE–Sec17–Sly1 co-complexes were formed at 30°C for 60 min, washed in either 30°C buffer or 4°C buffer, and incubated an additional 60 min at the same temperature as the prior wash. There was little Sly1 dissociation at 4°C and slightly more at 30°C (*Figure 8—figure supplement 1*), indicating that elevated temperature promotes productive SM–SNARE association rather than stabilizing extant SM–SNARE complexes. At 90% saturating levels, Sly1 bound the Sec17–SNARE complex with a half-time of ~20 min. Under similar conditions, Sly1 binding to the Sed5 N-peptide and Sec17 binding to Golgi SNARE complexes were both complete within 1 min (*Figure 8E*). When Sly1 was loaded onto SNARE–Sec17 complexes at 24° instead of 30°C, the time required to reach steady-state binding increased from 1 to 4 hr (*Figure 8F*). These observations indicate that the temperature dependence of SM binding is due to rate-limiting steps in SM–SNARE association rather than dissociation. A possible interpretation is that Vps33 and Sly1 slowly interconvert between ground states unable to bind SNARE complexes and activated, binding-competent, states. SNARE-bound Sec17—and perhaps other docking factors—would then elicit or stabilize activated SM conformations to accelerate SM loading onto cognate SNARE complexes. Because pre-incubation of Sly1 at 30°C did not accelerate subsequent Sly1 binding to Sec17-SNARE complexes (*Figure 8—figure supplement 2*), we suggest that elevated temperature increases the frequency of interconversion between the putative ground and binding-active conformations, rather than by stabilizing an activated conformation.

## Discussion

### Crosstalk between SNARE assembly and disassembly factors

Pioneering biochemical and genetic studies led to the idea that parallel N-to-C zippering of SNARE complexes *in trans* pulls membranes together to initiate fusion (*Hanson et al., 1997*; *Nichols et al., 1997*; *Poirier et al., 1998*; *Sutton et al., 1998*). SNAREs are sufficient to drive basal fusion of liposomes (*Figure 9A*, reaction i) and impose a layer of compartmental specificity (*Weber et al., 1998*; *McNew et al., 2000*). In these minimal systems, purified SM proteins accelerate fusion (reaction ii; *Scott et al., 2004*; *Shen et al., 2007*; *Furukawa and Mima, 2014*). However, the significant basal activities of SNAREs and the relatively modest rate enhancements conferred by SMs have been difficult to reconcile with absolute requirements for SM function in vivo.

Sec17, Sec18, and ATP completely suppress basal SNARE-mediated liposome fusion (*Figure 9A*, reaction iii), likely by binding and prematurely disassembling *trans*-SNARE complexes or their precursors

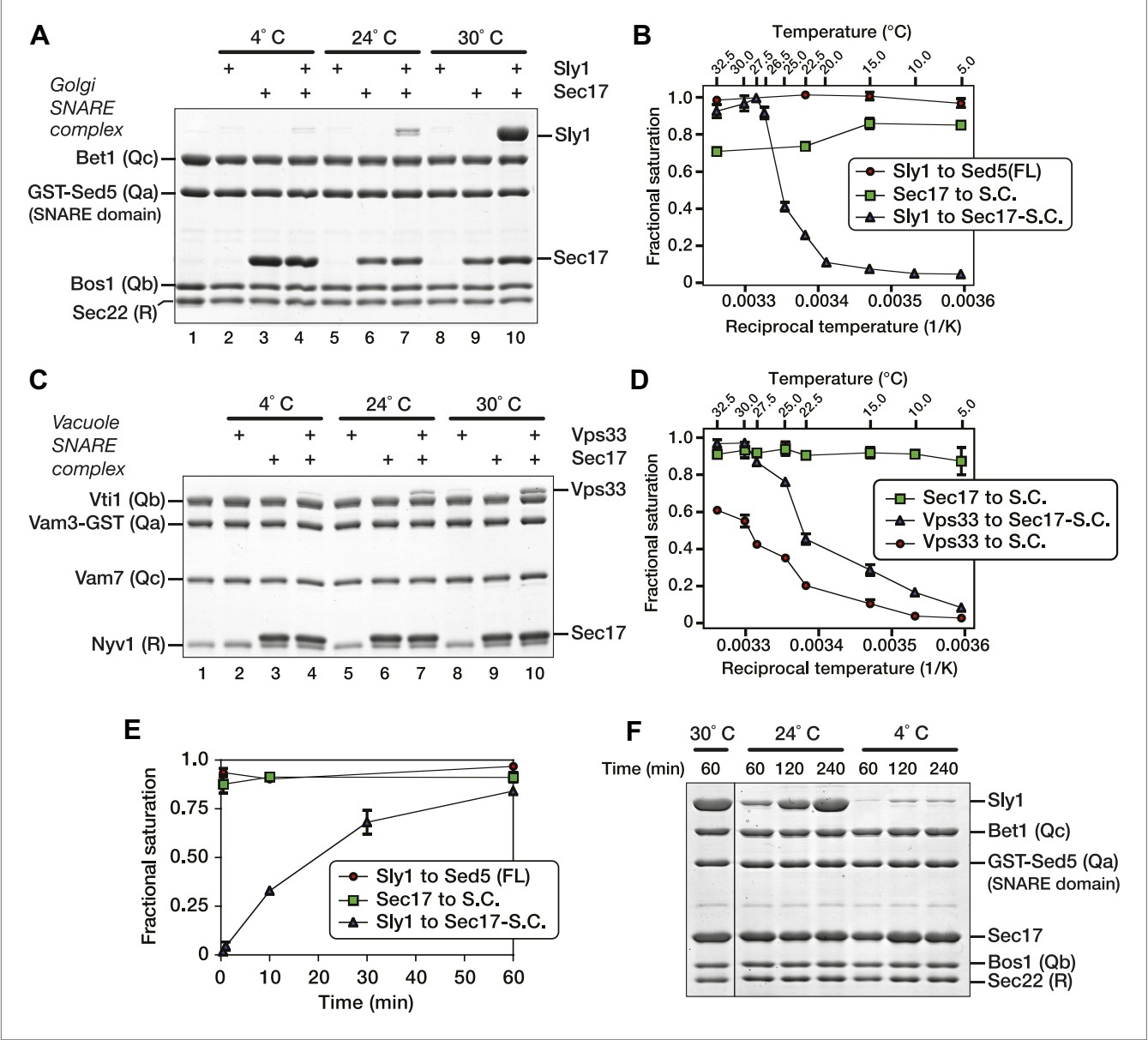

**Figure 8**. Thermal dependence of SM-SNARE complex association. (**A**) Golgi SNARE complexes were assembled on immobilized Sed5 SNARE domain (lacking the Sed5 N-peptide), then assayed for Sly1 (12 µM) binding in the absence and presence of Sec17 (12 µM). The binding reactions were incubated for 60 min at 4°, 24° or 30°C. (**B**) Sly1 binding to Golgi SNARE complexes (S.C.) was evaluated across a range of temperatures. Note that reciprocal temperature is plotted in units of 1/K, with warmer temperatures on the left side of the plot. Immobilized SNARE complexes (500 nM) were assayed for binding of sub-saturating amounts of Sly1 (8 µM, in the presence of 20 µM Sec17), or for binding of Sec17 alone (7 µM). In an additional control, Sly1 (6 µM) was assayed for binding to the N-peptide of Sed5 (FL; full-length cytoplasmic domain; 500 nM). Bound material was separated by SDS-PAGE, visualized with SYPRO Ruby, and quantified using standard curves of purified proteins of known concentrations. (**C**) The thermal dependence of Vps33–SNARE association was assayed as in **A**, except that Vps33 was present at 1 µM and Sec17 was present at 2 µM. (**D**) Vps33 binding to SNARE complexes was evaluated across a range of temperatures, similar to **B**. As indicated, Vps33 was present at 1.5 µM and Sec17 was present at 20 µM. (**E**) The kinetics of Sly1 binding at 30°C to Sec17–SNARE complex were analyzed and quantified using conditions and protein concentrations as in panel **B**. (**F**) Sly1 association kinetics are controlled by temperature. Immobilized Golgi SNARE complex (500 nM) was incubated with Sly1 (10 µM) and Sec17 (20 µM) at the indicated temperatures for the indicated times. Bound proteins were separated by SDS-PAGE and stained with Coomassie blue.

The following figure supplements are available for figure 8:

**Figure supplement 1**. Stability of Sly1-Sec17-SNARE complexes at 4° and 30°C.

**Figure supplement 2**. Pre-incubation at 30°C does not trigger conversion of Sly1 into a persistently activated state.

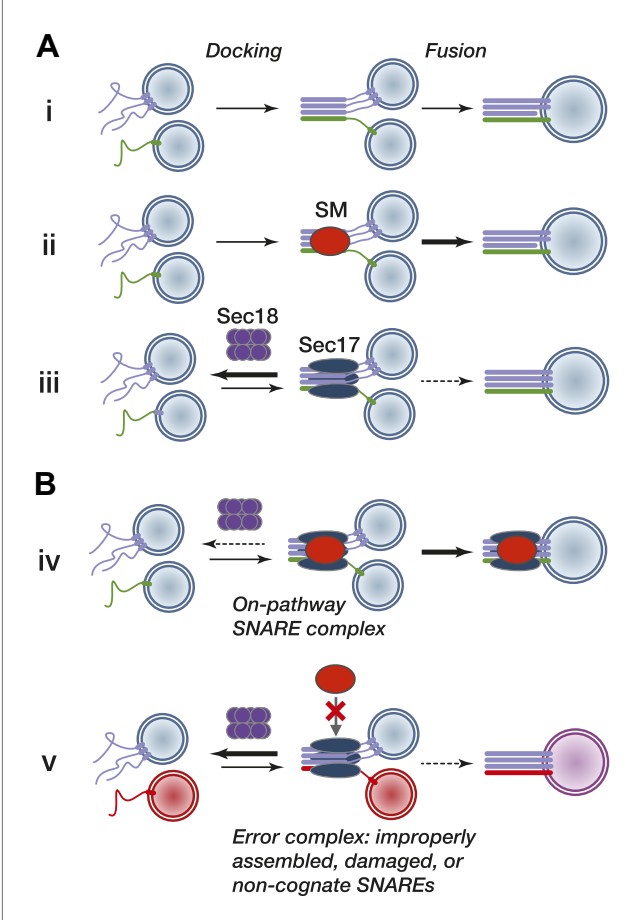

**Figure 9**. Working model. (**A**) Subreactions of SNARE-driven fusion. (**i**) Basal fusion, as with SNARE proteoliposomes. (**ii**) SM stimulation of the forward basal fusion reaction. Note that the SM may stimulate *trans*-complex assembly, the fusogenic activity of extant complexes, or both. (**iii**) Disassembly of nascent pre-fusion complexes by Sec17 and Sec18 impairs fusion. (**B**) SM stimulation of fusion in vivo. (**iv**) Sec17 accelerates SM loading onto cognate SNARE complexes, resulting in more efficient fusion and shielding of the complex from premature disassembly by Sec18. The location of the SM on the SNARE–Sec17 complex, and the dissociation of the SM from the post-fusion complex, are speculative. (**v**) The SM does not efficiently bind an improperly assembled, damaged, or non-cognate SNARE complex, exposing the complex to kinetic proofreading by Sec17 and Sec18.

(*Rohde et al., 2003*; *Starai et al., 2008*; *Ungermann et al., 1998*; *Xu et al., 2010*; but see also; *Weber et al., 2000*). In these systems the SM, in combination with other factors including HOPS, Munc13, and synaptotagmin, facilitates efficient fusion in the presence of Sec17 and Sec18, suggesting that SMs have a minimum of two fusion-promoting functions. First, SMs promote the assembly of *trans*-SNARE complexes or make them more fusogenic once assembled. Second, SMs protect *trans* complexes from premature disassembly by Sec18 (*Figure 9B*, reaction iv). Because multiple cofactors were present in previous studies, it was unclear whether SNARE complex protection from disassembly was attributable directly to SMs, or a property emergent from multiple proteins (*Starai et al., 2008*; *Ohya et al., 2009*; *Xu et al., 2010*; *Ma et al., 2013*). Our experiments now show that SMs functionally interact with the SNARE disassembly machinery in vivo, and affirm that SMs from at least two of the four subfamilies are sufficient to decrease rates of Sec18-mediated SNARE disassembly in vitro. Contrary to expectations, however, Vps33 and Sly1 did not compete with Sec17 for SNARE complex binding. Instead, SNARE-bound Sec17 accelerated SM loading, resulting in the efficient formation of SM–Sec17–SNARE co-complexes with 1:3:1 stoichiometry. Sec17 is therefore a multifunctional SNARE complex adapter, capable of recruiting not only the disassembly ATPase Sec18 but also compartment-specific SMs that oppose Sec18-mediated SNARE disassembly.

## SMs, Sec17, and the kinetics of SNARE complex persistence and activity

Our results and other converging lines of experimentation suggest a working model in which Sec17 binding to pre-fusion complexes leads to alternative fates (*Figure 9B*). When an SM recognizes and binds a Sec17–SNARE complex (reaction iv), it promotes fusion in at least three ways. First, the bound SM directly impedes SNARE disassembly by Sec18. Second, the SM augments the SNARE complex's fusogenic activity. Third, by accelerating fusion, the SM shortens the pre-fusion complex's lifetime, thereby shortening its temporal exposure to disassembly by Sec18 (kinetic partitioning; *Hardy and Randall, 1991*). In this view of fusion, SMs function as true enzymes: they bind and stabilize on-pathway intermediates, protect these intermediates from side reactions, and accelerate the conversion of inter-mediates to end products. Crucially, this model predicts the annihilation of pathway-specific fusion when SMs are deleted from living cells—a result that holds across many different organisms and fusion pathways. When SM function is compromised, pre-fusion SNARE complexes should be less fusogenic, persist for longer times, and be more exposed to premature disassembly by Sec17 and Sec18 (*Figure 9B*, reaction v).

A further implication of our working model is that SNAREs, Sec17, SMs, and Sec18 have precisely the features required to implement a kinetic proofreading system (*Hopfield, 1974*). Kinetic proofreading entails a sequence of independent, driven discrimination events. For SNARE-mediated fusion in vivo, there would be a minimum of two discrimination events. First, the nucleation of the *trans*-SNARE complex (an intermediate analogous to a Michaelis complex) has significant but not absolute intrinsic selectivity, derived mainly from packing interactions at the core of the SNARE bundle. Second, on-pathway SNARE bundles are positively selected by SM proteins. Sec17 detects the general shape and charge distribution of assembled SNARE bundles (*Marz et al., 2003*), while SMs identify pathway-specific SNARE configurations, protecting them from Sec18 and augmenting their forward fusogenic activity. Error products—off-pathway, incorrectly assembled, or damaged SNARE complexes—would not be efficiently bound by SMs (*Figure 9B*, reaction v). These complexes would fuse more slowly or not at all, and would be fully exposed to Sec17 and Sec18-mediated disassembly.

Although SNAREs selectively drive fusion when complexed with cognate partners (*McNew et al., 2000*), individual SNAREs readily enter into non-cognate, off-pathway, antiparallel or mis-registered complexes with substantial thermal stability and commensurately low off-rates (*Brunger et al., 2009*; *Furukawa and Mima, 2014*). Moreover, certain R-SNAREs (Nyv1, Snc2, Sec22) drive fusion with non-cognate Q-SNAREs (*McNew et al., 2000*; *Izawa et al., 2012*). Compartment-specific tethers confer additional selectivity by accelerating the forward rate of cognate SNARE pairing, but this may not be sufficient. Normal cellular transactions are replete with opportunities to assemble erroneous SNARE complexes that could drive inappropriate fusion or trigger unregulated and irreversible organelle aggregation. For example, mammalian endosomes progressively associate with endoplasmic reticulum (ER) until 98% of late endosomes and lysosomes are adjacent to ER, generally within 30 nm or less (*Friedman et al., 2013*). Proofreading would emplace a last line of defense against SNARE assembly errors and consequent defects in cell architecture and function.

## Additional considerations

We emphasize that our working schema is necessarily simplified and that pathway-specific specializations are likely to occur. At neuronal synapses, α-SNAP (Sec17) and the $Ca^{2+}$ sensor synaptotagmin compete for SNARE complex binding (*Sollner et al., 1993*). Within this specialized context (*Fasshauer et al., 1998*; *Jackson and Chapman, 2008*; *Südhof and Rothman, 2009*), synaptotagmin may interact with the SM (Munc18-1) to protect pre-fusion complexes in a manner analogous to Sec17 and Vps33 or Sly1.

The complete sequence of SNARE assembly events during priming, docking, and *trans* complex assembly has not been definitively established for any in vivo pathway. Off-pathway *cis*-SNARE complexes (e.g., Qa₂-Qb-Qc; *Fasshauer and Margittai, 2004*; *Fasshauer et al., 1997*; *Margittai et al., 2001*; *Pobbati et al., 2006*), undergo futile cycles of assembly and Sec18/NSF-mediated disassembly, and it has been suggested that SMs might positively select activated pre-docking intermediates (*Carr et al., 1999*; *Carr and Rizo, 2010*; *Kramer and Ungermann, 2011*; *Ma et al., 2013*; *Furukawa and Mima, 2014*). In this context it is notable that high levels of Vps33 and Sly1 remain associated with Qa-SNARE domains following complex disassembly by Sec17 and Sec18 (*Figure 3*), raising the possibility that SNARE disassembly is coupled to the formation of SM-SNARE subcomplexes prior to docking.

The biochemical experiments in this study were done with complexes in solution, free of membranes. However, the geometry of SNARE juxtamembrane domains in *cis* or *trans*, and especially the geometries of the membrane surfaces before and after fusion, may control whether Sec17 and SMs synergize or compete for binding. Consistent with this idea, Sec17 competes with the HOPS complex for binding to post-fusion *cis*-SNARE complexes on the yeast vacuole, as shown in meticulous co-isolation experiments (*Collins et al., 2005*). On the other hand, Sec17 interacts cooperatively with poised, partially-zipped *trans*-SNARE complexes on docked vacuoles, triggering fusion (*Schwartz and Merz, 2009*). This Sec17-dependent fusion requires Rab signaling and the HOPS effector complex—including the SM Vps33—but is totally independent of Sec18 and ATP (*Schwartz and Merz, 2009*).

Vps33 and Sly1 binding to core SNARE bundles, with or without Sec17, is slow and rate-limited by temperature (*Figure 8*), implying that significant conformational transitions are required for assembly of SM–SNARE complexes. The requirement for physiological (warm) temperatures in our in vitro assays may explain in part why SM–Sec17–SNARE interactions were not previously detected. In vivo, the relevant transitions may occur in a concerted manner; alternatively, they may involve ratchet-like sequential SM association with a series of SNARE assembly intermediates, with each sub-step traversing a smaller energy barrier. The central cavities of SMs are proposed binding sites for SNARE helical bundles and vary in the sizes of their openings, implying conformational flexibility (*Bracher and Weissenhorn, 2001*; *Bar-On et al., 2011*; *Baker et al., 2013*). Our observations provide new empirical support for models in which SM proteins and perhaps SNAREs must undergo substantial conformational transitions to productively associate. The nature of these transitions, and the specific biochemical events that allow SM binding and on-pathway fusion to occur over physiologically relevant time scales, remain to be elucidated.

## Materials and methods

### Plasmids

SNAREs and SM open reading frames were cloned into bacterial expresssion vectors using T4 DNA ligase (New England Biolabs, Beverly, MA) as previously described (*Lobingier and Merz, 2012*). In all cases, constructs lacked the SNARE transmembrane domains. In brief, the cytoplasmic domain of Vam3 (aa1-264) was cloned into NcoI/SacI-digested pRSF-1b with no N-terminal tag and C-terminal GST separated from the SNARE by a TEV cleavage site. N-terminal GST tags for the full cytoplasmic domain of Sed5 (aa1-319), or a Sed5$_{SNARE\ domain}$ (aa170-319) construct lacking the N-peptide and Habc domain, were cloned into BamHI/XhoI-cut pGST-Parallel1. The soluble domains of SNAREs were cloned as N-terminal His$_6$-tagged fusions into pHIS-Parallel1: Bos1 (aa1-222), Sec22 (aa1-188) were ligated into BamHI/XhoI-cut vector, while Nyv1 (aa1-231) was cloned into NcoI/SacI-digested vector. The SNARE domain of Vam7 (aa190-316) was inserted in-frame into a BamHI/PstI-cut His$_6$-GFP-TEV sequence. The soluble domains of Bet1 (aa1-123) and Vti1 (aa1-194) were cloned into BamHI/XhoI-cut or NcoI/PstI-cut (respectively) pRSF-1b carrying an N-terminal His$_7$-MBP tag. Full-length Sly1 (aa1-667) was cloned for expression in NcoI/SacI-digested pHIS-Parallel1. Vps33 was cloned for expression in the baculovirus system as described (*Brett et al., 2008*; *Lobingier and Merz, 2012*). Plasmids overexpressing SNARE disassembly machinery in yeast were made by gap repair recombination of PCR products containing *SEC17* and/or *SEC18*, each with 500 bases of promoter sequence and 300 bases of terminator sequence, into high copy (2μ) vectors pDN526 or pDN524 at unique SacI and HindIII restriction sites, respectively (see *Table 2*). We verified overexpression constructs by DNA sequencing and Western blotting of yeast lysates.

### Protein expression

Vps33 was expressed and purified from insect cells using the *Baculovirus* system as described (*Lobingier and Merz, 2012*). All other proteins were expressed in *E. coli* that harbored a pRIL codon-bias correction plasmid with the exception of Bos1, which was expressed in Rosetta2 pLys cells. Cells were inoculated at 0.05 OD$_{600}$, grown to 1.0–1.2 OD$_{600}$ in Terrific Broth, and expression was induced with 100 μM IPTG for overnight expression at 21°C (His$_6$-Sly1, His$_6$-Nyv1 and His$_7$-MBP-Vti1, GST-Sed5$_{SNARE\ domain}$, His$_6$-Bos1, His$_7$-MBP-Bet1, His$_6$-Sec22), 500 μM IPTG for 4–5 hr at 30°C (His$_6$-GFP-Vam7$_{SNARE}$), or 1 mM IPTG for 3 hr at 37°C (Vam3-GST and GST-Sed5).

### Protein purification

Cells expressing His-tagged proteins were lysed by sonication in Buffer A (50 mM HEPES, 200 mM NaCl, 10% [m/v] glycerol, 5 mM 2-meracptoethanol, 25 mM imidazole, 0.5% TritonX-100, pH 7.4)

**Table 2.** Yeast lines and plasmids employed in this study

| Name | Genotype | Reference or source |
| --- | --- | --- |
| *S. cerevisiae* | | |
| SEY6210 | *MATα leu2-3112 ura3-52 his3-200 trp1-901 lys2-801 suc2-9* | *Robinson et al., 1988* |
| WSY41 | SEY6210; *vps41Δ1::LEU2* | *Cowles et al., 1997* |
| BY4742 | *MATα his3Δ1 leu2Δ0 lys2Δ0 ura3Δ0* | ATCC |
| BLY3 | BY4742; *pep4Δ::KAN VPS33-ttx-GFP::NAT* | *Lobingier and Merz, 2012* |
| BLY5 | BY4742; *pep4Δ::KAN vps33 $^{R281A}$-ttx-GFP::NAT* | *Lobingier and Merz, 2012* |
| BLY6 | BY4742; *pep4Δ::KAN vps33car $^{[G297V]}$-ttx-GFP::NAT* | *Lobingier and Merz, 2012* |
| CBY267 | S288C; *MATα ade2-1 ura3-1 trp1-1 leu2-3112 can1-100* | *Cao et al., 1998* |
| RSY268 (CBY268) | S288C; *MATα ade2-1 ura3-1 trp1-1 leu2-3112 can1-100 sly1$^{ts}$* | *Cao et al., 1998* |
| Plasmids | | |
| pDN526 | Ap$^R$ *2µ URA3* | *Nickerson et al., 2012* |
| pDN313 | *SEC18* (pDN526) | This study |
| pDN314 | *SEC17* (pDN526) | This study |
| pDN315 | *SEC17 SEC18* (pDN526) | This study |
| pDN524 | Ap$^R$ *2µ TRP1* | This study |
| pDN316 | *SEC18* (pDN524) | This study |
| pDN317 | *SEC17* (pDN524) | This study |
| pDN318 | *SEC17 SEC18* (pDN524) | This study |
| pGO735 | Ap$^R$ *CEN LEU2 PGK1pr::RLuc SNA3-FLuc* (pRS415) | G Odorizzi (CU-Boulder) |
| pRP1 | pRSF Km$^R$ *His$_7$-MBP-(tev)-* | *Lobingier and Merz, 2012* |
| pBL14 | pBL12 Km$^R$ *VAM3 (1-264)-(tev)-GST* | *Lobingier and Merz, 2012* |
| pBL19 | pRP1 Km$^R$ *His$_7$-MBP-(tev)-VTI1 (1-194)* | *Lobingier and Merz, 2012* |
| pBL20 | pHIS Parallel1 Ap$^R$ *His$_6$-(tev)-NYV1 (1-231)* | *Lobingier and Merz, 2012* |
| pBL22 | pBL12 Km$^R$ *His$_6$-GFP$_{A207K}$-(tev)-Vam7 (190-316)* | *Lobingier and Merz, 2012* |
| pBL25 | pGST Parallel1 Ap$^R$ *GST-(tev)-SED5$_{SNARE}$ (170-319)* | This study |
| pBL26 | pHIS Parallel1 Ap$^R$ *His$_6$-(tev)-BOS1 (1-222)* | This study |
| pBL27 | pHIS Parallel1 Ap$^R$ *His$_6$-(tev)-SE22 (1-188)* | This study |
| pBL49 | pRP1 Km$^R$ *His$_7$-MBP-(tev)-Bet1 (1-123)* | This study |
| pBL50 | pGST Parallel1 Ap$^R$ *GST-(tev)-SED5$_{SNARE}$ (1-319)* | This study |
| pBL51 | pHIS Paralle1 Ap$^R$ *His$_6$-(tev)-Sly1* | This study |
| pSec17 | pTYB12 Ap$^R$ *CBD-(intein)-Sec17* | *Schwartz and Merz, 2009* |

supplemented with protease inhibitors. Cell lysates were clarified by centrifugation for 25 min at 18,500×*g* at 4°C. The supernatant was incubated with Ni-NTA HP resin (GE Heathcare, Piscataway, NJ) for 10 min at 4°C. The resins were washed extensively in Buffer A followed by washes in Buffer B (20 mM HEPES, 200 mM NaCl, 10% [m/v] glycerol, 2 mM 2-mercaptoethanol, 35 mM imidazole, pH 7.4). His-tagged proteins were eluted from the resin with Buffer B supplemented with 400 mM imidazole, and then exchanged into Storage Buffer (20 mM HEPES, 200 mM NaCl, 10% [m/v] glycerol, 2 mM 2-mercaptoethanol, pH 7.4) and snap-frozen in liquid nitrogen. Cells containing the GST-tagged SNAREs were lysed in Storage Buffer supplemented with protease inhibitors and 5 mM EDTA, and the clarified lysate was frozen in liquid nitrogen.

## Affinity assays

SNARE complexes were formed by binding 125 pmol of GST-SNARE (5.8 µg of GST-Sed5$_{SNARE domain}$ or 7.1 µg of Vam3-GST) to glutathione sepharose 4B resin (GE Healthcare) for 2 hr at 4°C. Resins were

washed twice with SM Assay Buffer: 20 mM HEPES, 150 mM NaCl, 2 mM 2-mercaptoethanol, 0.05% (m/v) Anapoe-X-100 (also called Triton-X-100; Affymetrix, Santa Clara, CA), pH 7.4. A ≥fivefold molar excess of Qb-, Qc-, and R-SNAREs was incubated overnight at 4°C with the GST-SNARE. For vacuole SNARE complexes, these were soluble domains of Vti1 and Nyv1 and the SNARE domain of Vam7: $His_7$-MBP-Vti1, $His_6$-GFP-$Vam7_{SNARE}$, and $His_6$-Nyv1. For Golgi SNARE complexes, these were soluble domains of Bos1, Bet1, and Sec22: $His_6$-Bos1, $His_7$-MBP-Bet1, $His_6$-Sec22. Unbound SNAREs were removed from SNARE complexes by washing the resins with SM assay buffer twice at 4°C and twice at room temperature. Sec17, the SM protein, or both were added at the indicated concentration to binding reactions containing immobilized SNARE complex (500 nM final, unless otherwise specified). Pulldowns were performed at 30°C for 1 hr, the resins were washed three times, and eluted with SM Assay Buffer supplemented with 20 mM reduced glutathione, pH 7.4. Samples were boiled in SDS-loading buffer, and separated using 12% SDS-PAGE for experiments using Sly1 or 10% SDS-PAGE for experiments using Vps33.

## SDS-PAGE imaging and quantification

Unless otherwise indicated, all gels shown were stained with Coomassie brilliant blue and imaged on an Epson 4490 transmission scanner. All experiments were repeated three times or more; representative gels are shown. Quantification of protein binding was performed using SYPRO-Ruby stain (Invitrogen, Carlsbad, CA) and a standard curve of each relevant protein, and gels were imaged using a Gel Doc XR+ (Bio-Rad, Hercules, CA). Where indicated, experiments with Vps33 were done twice rather than 3 or more times due to limitations in the amount of available protein. Data from three experiments (Sly1) or two experiments (Vps33) were plotted as fractional saturation of SM protein binding to immobilized SNARE complex, relative the total concentration of free SM protein in solution. $Kd_{obs}$ values and Hill coefficients were estimated by nonlinear fitting (GraphPad Prism v. 5) of a single-site binding model with Hill coefficient to the data. Two-site models did not substantially improve the quality of the fits. Thermal coefficients ($Q_{10}$) of SM association with SNARE or Sec17-SNARE complexes were calculated as

$$Q_{10} = \left(X_2/X_1\right)^{10/(T2-T1)}$$

where $X_1$ and $X_2$ are the binding efficiencies at lower and higher temperatures, $T_1$ and $T_2$ (*Hille, 2001*). The $Q_{10}$ values were extrapolated from the slope of the steepest part of the temperature-binding curve.

## SNARE disassembly and re-assembly

Sec18 activity was assayed in SNARE Disassembly Buffer, which was SM Assay Buffer with 1 mM ATP and 2 mM $MgCl_2$, pH 7.4. Unless otherwise noted, the SM protein and Sec17 were allowed to bind to SNARE complexes for 1 hr at 30°C before addition of Sec18. Disassembly reactions were then incubated for the indicated times and quenched by washing the samples in ice-cold SM Assay Buffer with 10 mM EDTA final, pH 7.4. Remaining resin-bound proteins were eluted with SM Assay Buffer containing 20 mM reduced glutathione, pH 7.4 at room temperature. To assay re-assembly of SNARE complexes, SNARE complexes were disassembled for 30 min at 30°C. Sec18 activity was then quenched with a final concentration of 10 mM EDTA, reactions were supplemented with additional soluble Qb, Qc, and R-SNAREs as indicated, and incubated at 30°C for 30 min.

## Cell growth and trafficking

Limiting plate dilutions were performed by growing yeast carrying plasmid vectors overnight at permissive temperature in synthetic media lacking Ura and with 2% (m/v) glucose and 0.05% (m/v) casamino acids. $OD_{600}$ was measured for each culture to permit equalization of cell mass. Cells were serially diluted onto synthetic media dropout plates. Growth curves in liquid were obtained as described (*Paulsel et al., 2013*) but used synthetic media lacking Ura and with 2% glucose and 0.05% casamino acids. To prevent cell clumping in the Bioscreen-C machine (Growth Curves USA), 0.2% Nonidet-P40 was added to synthetic media (*McIntosh et al., 2011*). For studies of $Zn^{2+}$ sensitivity, synthetic medium was prepared with asparagine at molar equivalence to, and in place of, the usual ammonium and adjusted to pH 6.5. This allowed 1 mM $ZnCl_2$ to remain soluble. YODA software (*Olsen et al., 2010*) was used to analyze growth curve data. Cargo protein sorting was assayed using LUCID (*Nickerson et al., 2012*).

## Acknowledgements

We thank Drs C Carr, E Chapman, S Gordon, J Mima, M Munson, J Rizo, C Stroupe, M von Zastrow, and members of the Merz lab for insightful discussions and penetrating comments on the manuscript. We thank C Barlowe and G Odorizzi for strains and reagents. Our work on membrane fusion is supported by NIH-NIGMS GM077349. AJM received additional support as a Research Scholar of the American Cancer Society (ACS), DPN was a Postdoctoral Fellow of the ACS, and BTL was partially supported by NIH-NIGMS T32 GM07270.

## Additional information

### Funding

| Funder | Grant reference number | Author |
| --- | --- | --- |
| National Institute of General Medical Sciences | RO1 GM077349 | Alexey J Merz |
| American Cancer Society | RSG-10-026-01-CSM | Alexey J Merz |
| National Institute of General Medical Sciences | T32 GM07270 | Braden T Lobingier |

The funders had no role in study design, data collection and interpretation, or the decision to submit the work for publication.

### Author contributions

BTL, DPN, Conception and design, Acquisition of data, Analysis and interpretation of data, Drafting or revising the article; S-YL, Acquisition of data, Analysis and interpretation of data, Drafting or revising the article; AJM, Conception and design, Analysis and interpretation of data, Drafting or revising the article

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
