## [Decision Letter]

Thank you for sending your work entitled “SM proteins Sly1 and Vps33 co-assemble with Sec17 and SNARE complexes to oppose SNARE disassembly by Sec18” for consideration at *eLife*. Your article has been favorably evaluated by a Senior editor and 3 reviewers, one of whom is a member of our Board of Reviewing Editors.

The Reviewing editor and the other reviewers discussed their comments before we reached this decision, and the Reviewing editor has assembled the following comments to help you prepare a revised submission.

All reviewers found your work interesting and recommended publication after appropriate revision. In particular, simultaneous binding of Sec17 and an SM protein to a SNARE complex is unprecedented and unexpected. By altering our fundamental understanding of how the key components of the membrane fusion machinery interact, this discovery seems likely to have a major impact on future work in the field.

There were several technical concerns and questions for clarification. Of these, there is one all reviewers would like you to address:

The findings from Carpp et al. (JCB 2006) that a point mutation in Vps45 can shift the equilibrium binding between a Tlg2 SNARE-binding mode to a SNARE complex-binding mode is consistent with the idea presented here that increasing the temperature increases the probability that the SM protein can shift into a SNARE complex-binding mode. Hence, the increased rate of binding at higher temperatures. Perhaps pre-heating the SM protein alone first would increase the rate of binding when subsequently added to Sec17-SNARE complexes (or SNARE complexes alone), this could indicate that the conformational change is restricted to the SM protein only. Other issues to address – note that we do not consider it as essential to carry out additional experiments to address them – they are for your consideration and/or editorial changes.

*Reviewer*
*#1:*

1) While the evidence for SM-Sec17-SNARE complexes seems overwhelming, I was not entirely convinced that assembly of this 'ternary' complex is “highly cooperative” Establishing cooperativity would seem to require additional evidence or, if I am missing something, at least a more careful marshalling of the existing evidence.

2) I can see why the authors favor the model that the SM protein binds to a composite surface that includes both Sec17 and the SNARE bundle. Nonetheless, I don't think that an alternative model can really be ruled out in which the SM protein recognizes a distinct configuration of Sec17 molecules achieved only when they are bound to the 'cognate' SNARE complex.

3) In relation to Figure 3, it's important to mention that syntaxin 5 lacks the N-terminal peptide. This becomes a major point later. Possibly, to avoid complicating the narrative, it would be OK to mention this only in the legend of Figure 3.

4) “The Sec17 concentration required to saturate Golgi SNARE complex was about four-fold greater than for vacuole SNARE complex.” Are these data not shown?

*Reviewer*
*#2:*

1) In contrast to the findings shown here, Togneri et al (PNAS 2006), demonstrated that (at least under the conditions tested) the yeast SM protein Sec1 was unable to prevent disassembly of the exocytic SNARE complexes by Sec17/18. This should be discussed; is it possible that Sec1 is sufficiently different from Sly1 and Vps33?

2) Along those lines, a more “universal” picture would be gained by demonstrating that other SM proteins also show these conserved features. Vps45 and Sec1 could be tested, to complete the yeast family, and/or Munc18-1 or Munc18c. This may be beyond the scope of this manuscript.

3) It is not clear, given the SNAP-SNARE complex binding data and model from Marz et al. (JBC 2003), and a variety of SM mutants with disrupted binding to SNARE complexes, how to build a model for the SM and 3 copies of Sec17 binding to SNARE complexes at the same time. There does not seem to be enough room available.

4) “...but Sec17 does not bind either Sly1 or Sed5 in the absence of the assembled SNARE complex.” This is misleading, because Figure 6 actually shows that Sec17 can bind Sly1, IF Sly1 is not bound to full-length Sed5 (Figure 6, not 6C). This is a curious finding that might indicate that binding of the N-peptide of Sed5 may affect the conformation of Sly1 (as suggested in Furgason et al (PNAS 2009) for Vps45-Tlg2 binding. In the cell, the syntaxin-family SNAREs will have their N-terminal domains and N-peptides, how does this affect the proposed model?

5) Use of the word “universal”, especially “universal co-chaperone” and “universal disassembly engine”. Although many of us would appreciate the membrane trafficking centric view of biology, this terminology, it is quite confusing due to many other co-chaperones and disassembly machine for other non-membrane trafficking biological processes.

6) The Q10 thermal coefficient needs to be better defined, referenced and the calculations described in M&M

7) In the figures, the axis need to be consistently clear whether the protein being tested is full-length Sec5 or the SNARE domain only (the labels change from panel to panel).

8) In Figure 3, it is curious that the SM stays bound to the SNARE complexes after disassembly by Sec18/Sec17, while Sec17 is removed. It is not clear how this fits into the overall picture of SM-Sec17 function proposed here.

9) Not necessary to provide the genotype for commercially available E. coli strains.

10) The plasmid information in Table 2 is rather limited and the relevant information is not obvious from the Materials and methods

*Reviewer*
*#3:*

1) The problem is that conclusions are drawn about pre-fusion complexes but all biochemical experiments are carried out with fully assembled SNARE complexes that represent the post-fusion state. To me it is unclear why SM proteins should block disassembly of a “cis” SNARE complex such as the one immobilized on the beads - biologically this does not make much sense. It needs to be explained how the SM proteins (or NSF) differentiate between prefusion and postfusion complexes (apparently the authors are aware of the problem – see question mark on the right in Figure 9, pathway IV). Thus the cartoon does not represent the findings correctly as all experiments were done with SNARE complexes shown on the far right in the cartoon (after fusion). Answering this question would require different approaches and is probably not straightforward.

2) I have some concerns about the fact that all experiments are carried out on beads under equilibrium conditions, i.e., using long incubation times, with the Q-a SNARE being immobilized. For instance, it is unclear to me why such long preincubation is required for the SM-proteins to exert their inhibitory effect on Sec18p-driven disassembly – as far as I remember previous work binding is fast. Similarly, I consider it as problematic to carry out re-assembly kinetics in such a manner – the bands are barely visible (Figure 4) and only become convincing after adding an approx. 6fold excess of soluble SNAREs, and kinetic resolution is poor. Furthermore, it is conceivable that binding of SM-proteins to a preassembled complex including SNAREs and alpha-SNAP may nonspecifically prevent access of the bulky Sec18p protein (that may not be able anymore to enter the pores of the resin) rather than blocking access to the SNARE complex. For these reasons I suggest to carry out at least one or two control experiments in solution or with the Qa-SNARE being incorporated into the membrane of liposomes.

---

## [Author Response]

*The findings from Carpp et al. (JCB 2006) that a point mutation in Vps45 can shift the equilibrium binding between a Tlg2 SNARE-binding mode to a SNARE complex-binding mode is consistent with the idea presented here that increasing the temperature increases the probability that the SM protein can shift into a SNARE complex-binding mode. Hence, the increased rate of binding at higher temperatures. Perhaps pre-heating the SM protein alone first would increase the rate of binding when subsequently added to Sec17-SNARE complexes (or SNARE complexes alone), this could indicate that the conformational change is restricted to the SM protein only*.

This is an important question. We performed the suggested experiment (now included as Figure 8—figure supplement 2). The results show that pre-incubation of Sly1 at 30° C versus lower temperatures has no effect on the subsequent efficiency of Sly1 binding to the SNARE-Sec17 complex. This indicates that elevated temperature does not induce a time-stable binding competence to Sly1 or the other proteins in the system, and is consistent with the interpretation that binding requires Sly1 or perhaps another molecule in the system to equilibrate between two states: a sparsely-occupied (higher-energy), binding-competent state, and a densely-occupied (lower-energy), binding incompetent state. Elevated temperature would increase the rate of inter-conversion between these states, elevating the rate of productive complex formation.

Reviewer #1:

*1) While the evidence for SM-Sec17-SNARE complexes seems overwhelming, I was not entirely convinced that assembly of this 'ternary' complex is “highly cooperative” Establishing cooperativity would seem to require additional evidence or, if I am missing something, at least a more careful marshalling of the existing evidence*.

The relevant distinction may be semantic, but that does not mean it is unimportant. There are multiple types of cooperativity, among which one may delineate at least two classical cases (Ptashne, 2007). The first case is positive conformational cooperativity (allostery), exemplified by the affinity increase that occurs as molecules of O2 sequentially bind hemoglobin. The second case is simpler and more general: the combined effect of multiple binding sites (avidity). It is exemplified by binding of monomeric phage λ repressor to its individual DNA operator half-site, versus the considerably stronger binding of λ repressor to its native palindromic operator, resulting from a low-affinity dimerization interface on the repressor protein. This type of cooperativity can in principle occur solely through rigid body interactions, or it can be coupled to conformational dynamics. Although we do not yet have a deep mechanistic understanding of how SM-Sec17-SNARE assembly occurs, the present experiments strongly support this more general kind of cooperative assembly. We removed the word “highly” to more precisely convey what can, and cannot yet, be said about the mechanisms that drive this assembly reaction.

*2) I can see why the authors favor the model that the SM protein binds to a composite surface that includes both Sec17 and the SNARE bundle. Nonetheless, I don't think that an alternative model can really be ruled out in which the SM protein recognizes a distinct configuration of Sec17 molecules achieved only when they are bound to the 'cognate' SNARE complex*.

For an SM to exhibit selectivity for its cognate SNARE-Sec17 complex without direct SM-SNARE contacts, Sec17 would need to bind each SNARE complex in a distinct conformation, and these conformational differences would then need to be transmitted to, and recognized by, the SM. This is far from the simplest explanation of our results but it cannot at present be excluded. We note that Vps33 has the intrinsic ability to recognize cognate versus non-cognate SNARE complexes. That is, Sec17-dependent and Sec17-independent binding of Vps33 to SNARE complex are of different affinities, but exhibit comparable selectivity (Figure 7, compare Vps33 binding in lanes 2 and 2´; see also [43]. We have modified the Results to emphasize the selectivity of this recognition rather than its underlying mechanism.

*3) In relation to*
Figure 3*, it's important to mention that syntaxin 5 lacks the N-terminal peptide. This becomes a major point later. Possibly, to avoid complicating the narrative, it would be OK to mention this only in the legend of*
Figure 3.

We agree and have made the suggested changes. We emphasize that in the SNARE disassembly assays, comparable results were obtained with SNARE complexes built on the Sed5 SNARE domain (Figure 3), or with complexes built on the full Sed5 cytoplasmic domain (Figure 3—figure supplement 3).

*4) “The Sec17 concentration required to saturate Golgi SNARE complex was about four-fold greater than for vacuole SNARE complex.” Are these data not shown*?

The divergent Sec17 affinity for different SNARE complexes was initially noted in binding optimization studies that are not shown. However, the difference is evident in experiments that are shown, as now noted in Results. Compare the amount of Sec17 required to saturate vacuole SNARE complex (5 µM; Figure 5) vs. Golgi SNARE complex (20 µM; Figure 6). The reported dissociation constant (half-maximal saturation) of α-SNAP for neuronal SNARE complexes in solution is ∼5 µM ([46]; Winter et al., 2009).

Reviewer #2:

*1) In contrast to the findings shown here, Togneri et al (PNAS 2006), demonstrated that (at least under the conditions tested) the yeast SM protein Sec1 was unable to prevent disassembly of the exocytic SNARE complexes by Sec17/18. This should be discussed; is it possible that Sec1*
*is sufficiently different from Sly1 and Vps33?*

The referee is correct. Indeed, like Togneri et al. for Sec1, [58], reported that Sly1 does not prevent NSF-mediated SNARE disassembly. However, in these studies only endpoints and not intermediate stages of disassembly were examined. In the present experiments disassembly also ran to completion, but in the presence of bound Vps33 or Sly, we observed kinetic delays (vs. absolute blockades of disassembly) at intermediate time points.

*2) Along those lines, a more “universal” picture would be gained by demonstrating that other SM proteins also show these conserved features. Vps45 and Sec1 could be tested, to complete the yeast family, and/or Munc18-1 or Munc18c. This may be beyond the scope of this manuscript*.

Agreed. We see the comparative analysis of Vps33 and Sly1 as a strength of the present experiments, and anticipate that the results will spur additional experimentation from other scientists working on SM proteins.

*3) It is not clear, given the SNAP-SNARE complex binding data and model from Marz et al. (JBC 2003), and a variety of SM mutants with disrupted binding to SNARE complexes, how to build a model for the SM and 3 copies of Sec17 binding to SNARE complexes at the same time. There does not seem to be enough room available*.

This is a major question, particularly at the synapse where Munc-13, Munc-18, synaptotagmin, complexin, and perhaps other factors all appear to engage the pre-fusion trans-SNARE complex. At present, the most plausible model for how an SM might engage the tetrahelical SNARE bundle is derived from the crystal structure of Munc-18/nSec1 bound to the closed form of syntaxin, which folds into a tetrahelical bundle that might be a structurally analogous to the SNARE complex (Misura et al., 2000).Author response image 1.

If this structural analogy is broadly correct, well over half of the SNARE bundle circumference, and most of the bundle’s surface area, should be exposed for Sec17/α-SNAP binding. In a recent single-particle EM study of 20S SNARE- α-SNAP-NSF complexes (Chang et al., 2012), C3 (threefold rotational) symmetry was imposed for SNARE-bound Sec17, resulting in a C4:C3 SNARE–α-SNAP pseudosymmetry mismatch. To the best of our knowledge there is no experimental basis for this imposition of C3 symmetry. In an alternative model, Hanson’s group proposed that three adjacent Sec17/α-SNAP molecules wrap around one side of the SNARE complex (46) – an arrangement that could readily accommodate three Sec17/α-SNAP molecules and one SM per SNARE bundle, as observed in our experiments.

*4) “...but Sec17 does not bind either Sly1 or Sed5 in the absence of the assembled SNARE complex.” This is misleading, because*
Figure 6
*actually shows that Sec17 can bind Sly1, IF Sly1 is not bound to full-length Sed5 (*Figure 6*, not 6C). This is a curious finding that might indicate that binding of the N-peptide of Sed5 may affect the conformation of Sly1 (as suggested in Furgason et al (PNAS 2009) for Vps45-Tlg2 binding. In the cell, the syntaxin-family SNAREs will have their N-terminal domains and N-peptides, how does this*
*affect the proposed model?*

There are a few issues to discuss here. First, the interaction between the Sly1 and the N-peptide of Sed5 is not required for fusion (59). Vps33 does not have an N-peptide binding pocket and does not interact with the N-peptide of Vam3. Moreover, Vam3 does not enter into a closed conformation and the N-terminal domains of Vam3 are nonessential for Vam3 function (Dulubova et al., 2001; [43]; [1]; Wang et al., 2001; Laage and Ungermann, 2001). Similarly, yeast Sec1 does not interact with Qa-SNARE N-peptide (77). Thus, interactions between SM proteins and Qa-SNARE N-peptides are not universal and, where they do occur, they are not consistently important for SM function. If there is indeed a general function of SM proteins in SNARE-mediated fusion, that function cannot generally depend on Qa-SNARE N-peptide interactions. Nevertheless, it is still possible that Sed5 N-peptide binding to Sly1 initiates a conformational change that regulates Sly1 assembly dynamics, as suggested for Vps45 and for mammalian Sly1 (Furgason et al., 2009; Arac et al., 2005).

Figure 6 may indeed indicate some affinity between Sly1 and Sec17. However, the main conclusion of this experiment is that Sly1 does not readily bind the SNARE core bundle unless Sec17 is present. Because this experiment was analyzed by immunoblot, we caution that additional control experiments would be required before we could definitively conclude that the Sec17 signal observed in the absence of SNARE complex (Figure 6, lane 2) is specific binding rather than background. This is particularly so given the result of Figure 6 (lanes 7 and 8), where Sec17 binding to Sed5-tethered Sly1 is not detected. We have modified the relevant passage in the Results to clarify the nature of this uncertainty.

*5) Use of the word “universal”, especially “universal co-chaperone” and “universal disassembly engine”. Although many of us would appreciate the membrane trafficking centric view of biology, this terminology, it is quite confusing due to many other co-chaperones and disassembly machine for other non-membrane trafficking biological processes*.

We've deleted the word “universal,” and taken some care to specify that Sec17 and Sec18 are general SNARE disassembly factors rather than universal protein chaperones.

6) The Q10 thermal coefficient needs to be better defined, referenced and the calculations described in M&M

Estimation of Q10 is now described in the Methods, and a reference to classical applications is cited (Hille, 1991, p. 51 ff).

*7) In the figures, the axis need to be consistently clear whether the protein being tested is full-length Sec5 or the SNARE domain only (the labels*
*change from panel to panel)*

Done.

*8) In*
Figure 3*, it is curious that the SM stays bound to the SNARE complexes after disassembly by Sec18/Sec17, while Sec17 is removed. It is not clear how this fits into the overall picture of SM-Sec17 function proposed here*.

We agree. Additional in vitro and in vivo experimentation will be needed to evaluate the significance of this observation.

*9) Not necessary to provide the genotype for commercially available E. coli strains*.

OK.

*10) The plasmid information in*
Table 2
*is rather limited and the relevant information is not obvious from the Materials and methods*

We have more clearly described how the plasmids were constructed.

Reviewer #3:

*1) The problem is that conclusions are drawn about pre-fusion complexes but all biochemical experiments are carried out with fully assembled SNARE complexes that represent the post-fusion state. To me it is unclear why SM proteins should block disassembly of a “cis” SNARE complex such as the one immobilized on the beads – biologically this does not make much sense. It needs to be explained how the SM proteins (or NSF) differentiate between prefusion and postfusion complexes (apparently the authors are aware of the problem – see question mark on the right in*
Figure 9*, pathway IV). Thus the cartoon does not represent the findings correctly as all experiments were done with SNARE complexes shown on the far right in the cartoon (after fusion). Answering this question would require different approaches and is probably not straightforward*.

There are two critical differences between pre-fusion trans-SNARE complexes and post-fusion cis-SNARE complexes: first, differences in the conformations of the C-terminal (juxtamembrane) segments of the proteins; and second, differences in the topology and topography of the membranes within which the SNAREs are anchored. With only cytoplasmic domains (but not transmembrane segments and the associated membranes), it is inappropriate to identify the complexes studied here as either cis or trans.

In previous work we demonstrated that authentic kinetically stable trans-SNARE complexes assembled through the native docking pathway can be trapped on intact vacuolar lysosomes. These trans-complexes are zipped to packing layer +3 or +4; we and others have presented strong evidence that Sec17 can functionally and physically interact with these trapped trans-complexes (68; 88). Moreover, single-molecule laser tweezers experiments with neuronal SNARE complexes are fully consistent with the interpretation that pre-fusion complexes zipper to layer +3 or +4 (Gao et al., 2012). Thus, ∼70% of the SNARE bundle can be fully folded in the pre-fusion trans-complex.

The SM binding footprint on the SNARE bundle has not been mapped in detail for any SM. Consequently, it is still unclear whether Sec17, SMs, or both operating in concert, can discriminate in functionally meaningful ways between cis- and trans-SNARE complexes. However, we emphasize that functional experimentation with liposomes initially prompted the hypothesis that SM proteins protect trans-SNARE complexes from premature disassembly ([44]; Mima and Wickner, 2009; [50]; [88]). Our preferred working model focuses on SM and Sec17 association with pre- versus post-fusion complexes, because that interpretation is most consistent not only with the present in vivo and biochemical experiments, but with functional studies published by other laboratories. If our working model inspires experiments that cause it to be supplanted by a better model, it will have served its purpose.

*2) I have some concerns about the fact that all experiments are carried out on beads under equilibrium conditions, i.e., using long incubation times, with the Q-a SNARE being immobilized. For instance, it is unclear to me why such long preincubation is required for the SM-proteins to exert their inhibitory effect on Sec18p-driven disassembly – as far as I remember previous work binding is fast*.

To our knowledge, no one has previously reported Sly1 binding directly to the SNARE core bundle. In previous work, SNARE complexes were assembled on the full cytoplasmic domain of Sed5 – including the Sed5 N-peptide, which Sly1 binds with low-nM affinity. Because the on-rate for this high-affinity binding interaction is fast (Figure 8), the slower on-rate of Sly1 binding to the SNARE core bundle would have been obscured by the high-affinity Sly1-N-peptide interaction in previously published experiments. The slow on-rate of Sly1 (and of Vps33) binding to SNARE core bundles could reflect a requirement for another protein or a small molecule that accelerates binding, or it could reflect the initial recruitment of the SM to a SNARE assembly intermediate. These issues were raised at length in the original Discussion. In the current submission, the text has been modified for greater clarity.

*Similarly, I consider it as problematic to carry out re-assembly kinetics in such a manner – the bands are barely visible (*Figure 4*) and only become convincing after adding an approx. 6fold excess of soluble SNAREs, and kinetic resolution is poor*.

The experiments in Figure 4 are controls, and are set up specifically not to show assembly end-points, but rather to ascertain whether the increased abundance of SNARE complexes following disassembly in the presence of Vps33 or Sly1 (Figure 3) reflect increases in re-assembly. The experiment was run under the conditions of Figure 3 specifically to control for that possibility. We reiterate that experiments showing that Sly1 and Vps33 do not stimulate SNARE assembly in solution have also been reported by other laboratories (31; 39; 58).

*Furthermore, it is conceivable that binding of SM-proteins to a preassembled complex including SNAREs and alpha-SNAP may nonspecifically prevent access of the bulky Sec18p protein (that may not be able anymore to enter the pores of the resin) rather than blocking access to the SNARE complex. For these reasons I suggest to carry out at least one or two control experiments in solution or with the Qa-SNARE being incorporated into the membrane of liposomes*.

The relatively low density of SNARE complex on the affinity resin, and the complete disassembly we observe, suggests that steric occlusion is not a major issue. Moreover, the multiple-turnover rate we observe for Sec18-mediated disassembly of Golgi SNARE complexes (1.35±0.01 complexes • Sec18 hexamer-1 • min-1; Figure 3—figure supplement 2) are identical to the rate observed in solution for NSF-mediated disassembly of neuronal complexes under similar ionic conditions (∼1 complex • Sec18 hexamer-1 • min-1; [13]). Moreover, the rate of disassembly in our assays is linear over the time intervals assayed, further indicating that there is not a less-accessible subpopulation of complexes buried within the affinity matrix.